# Geometry of Nash Mirror Dynamics: Adaptive $\beta$-Control for Stable and Bias-Robust Self-Improving LLM Agents

## Abstract

Self-improving agents learn by playing competitive, often non-transitive language games (e.g., generator–solver, proposer–verifier) where training can oscillate or drift toward undesirable behaviours. We study this scenario through the lens of reverse-KL regularised Nash learning, showing how the regularisation strength $\beta$ shapes both where agents converge and how they get there. We derive a continuous-time view of Nash Mirror Descent (Nash-MD), revealing a simple geometry: trajectories are spirals on the simplex whose damping grows with $\beta$, while $\beta$ simultaneously pulls equilibria toward the reference policy—amplifying any existing biases. We prove last-iterate convergence to the $\beta$-regularised Nash equilibrium, quantify its first-order shift from the unregularised solution, and link convergence speed to the spectrum of the linearised dynamics.

Building on this geometry, we introduce two adaptive $\beta$ controllers: (i) a Hessian-based rule that targets a desired damping–rotation ratio to accelerate without overshoot, and (ii) a bias-based rule that caps measurable bias (e.g., output length, calibration, hallucination proxies) while retaining speed. On toy games (e.g. Rock–Paper–Scissors) and small open-model reasoning benchmarks, our controllers deliver faster, more stable convergence with bounded bias, outperforming baselines. The result is a practical recipe: tune $\beta$ as a control knob to make self-improving LLM agents both faster and safer.

## 1 Introduction

Self-improving LLM (large language model) agents (Tao et al., 2024; Tian et al., 2024; Munos et al., 2024; Rosset et al., 2024; Choi et al., 2025; Huang et al., 2025a;b) are trained in *competitive, non-transitive* preference games (e.g., proposer–verifier, generator–solver). In such games, standard no-regret or mirror-descent updates can exhibit *rotational* components that cause cycling and unstable last-iterate behaviour even when time-averages converge (Shapley, 1963; Hofbauer, 1996; Mertikopoulos et al., 2018). In practice, reverse-KL regularisation to a reference policy $\mu$ is pervasive in post-training and preference-learning pipelines (PPO-style KL in RLHF; DPO/IPO's implicit anchoring; Nash learning with entropic regularisation), where it is believed to stabilise learning and bound distribution shift (Munos et al., 2024; Ye et al., 2024; Xiong et al., 2024; Rafailov et al., 2023; Xiong et al., 2024; Zhao et al., 2024; Wang et al., 2024). However, the role of the *reverse temperature* $\beta$ remains under-characterised: how exactly does $\beta$ control damping vs. rotation of the dynamics? How far does it bias equilibria toward $\mu$? And how should $\beta$ be *adapted online* to trade convergence speed against explicit *bias budgets* (e.g., output length, calibration)?

**Contributions.** Our technical and empirical contributions are:

C1. Derive entropic Nash mirror ODEs with reverse-KL regularization (§2.2).
C2. Show a spectral separation $\lambda(J_\beta) = \{-\beta \pm i\,\sigma_k\}$ that isolates damping (real part) from rotation (imaginary part) (§2.4).
C3. Characterise $\beta$-regularised Nash equilibria as logit/quantal responses around $\mu$ (§2.3).
C4. Prove an $O(\beta)$ first-order equilibrium sensitivity bound near the unregularised NE (§2.3).
C5. Propose two adaptive-$\beta$ controllers: *Hessian-$\beta$* (spectral damping ratio) and *Bias-$\beta$* (log-scale updates to meet bias budgets) (§3).

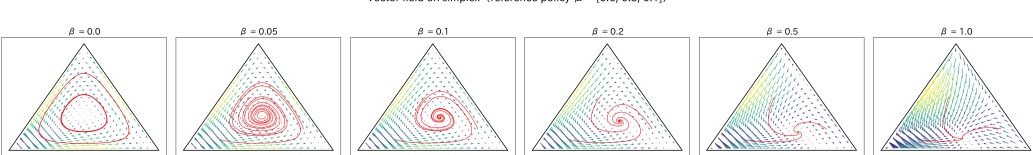

Figure 1: *RPS geometry under reverse–KL Nash mirror dynamics.* Simplex vector fields (RPS) for $\beta \in \{0, 0.05, 0.1, 0.2, 0.5, 1.0\}$; spiral steepening with $\beta$. Trajectory bundles from random inits; faster radial decay at larger $\beta$. *Takeaway:* $\beta$ adds uniform damping without changing rotational frequencies (Section 2).

C6. Validate on toy RPS (fields, trajectories, $\beta$-sweeps, spectra, convergence vs. baselines) and demonstrate feasibility on a small LLM micro-experiment (learned payoff) (§5).

**Hypotheses.** (A) Reverse-KL regularization ($\beta$) shifts fixed points away from the unregularised Nash equilibrium with a $\beta$-controlled bias; larger $\beta$ amplifies pre-existing reference-policy biases (length, overconfidence, hallucinations). (B) Convergence speed and spiral steepness increase with $\beta$. We formalise these as Theorems 2.4, 2.3, and 2.7.

**What is new vs. known.** Prior work studied mirror descent in games and regularised best responses. Our novelty is a *reverse-KL* geometry for Nash mirror dynamics that yields (i) a simple spectral law with additive $-\beta$ damping; (ii) a first-order sensitivity bound linking $\beta$ to equilibrium bias; and (iii) controllers that use this geometry to regulate convergence speed and explicit bias budgets.

## 2 THEORY: GEOMETRY OF REVERSE–KL NASH MIRROR DYNAMICS

We give a precise account of reverse–KL Nash mirror dynamics. Assumptions are explicit, theorem statements appear in the main text with concise proof sketches, and full proofs are in the Appendix.

### 2.1 PROBLEM SET-UP AND ASSUMPTIONS

Let $\Delta_m = \{p \in \mathbb{R}^m_{\geq 0} : \mathbf{1}^\top p = 1\}$ and $\mathsf{T} = \{v \in \mathbb{R}^m : \mathbf{1}^\top v = 0\}$. We study the reverse–KL regularised two-player zero-sum matrix game

$$f_\beta(x,y) = x^\top A y \ - \ \beta \, D_{\mathrm{KL}}(x\|\mu) \ + \ \beta \, D_{\mathrm{KL}}(y\|\mu), \qquad x, y \in \Delta_m, \ \beta \geq 0, \tag{1}$$

where $A \in \mathbb{R}^{m \times m}$ and $\mu \in \mathrm{relint}(\Delta_m)$ is the reference. We use the negative-entropy mirror map $\psi(p) = \sum_i p_i \log p_i$ with dual variables $z_x, z_y$ and $x = \mathrm{softmax}(z_x)$, $y = \mathrm{softmax}(z_y)$.

**Assumptions (local, minimal).**

**A0 (Reference positivity).** $\mu \in \mathrm{relint}(\Delta_m)$ (all coordinates strictly positive).

**A1 (Interior equilibrium).** For the $\beta$ considered, there exists $(x^*_\beta, y^*_\beta) \in \mathrm{relint}(\Delta_m)^2$ solving $\max_x \min_y f_\beta(x,y)$.

**A2 (Local variational stability).** The Nash field $F(x,y) = (\nabla_x f_\beta, -\nabla_y f_\beta)$ is locally monotone on $\mathsf{T} \times \mathsf{T}$ around $(x^*_\beta, y^*_\beta)$; when needed, strong monotonicity holds with modulus proportional to $\beta$.

**A3 (Smoothness and nonsingularity at $\beta{=}0$).** Gradients are $C^1$ near $(x^*_0, y^*_0)$; the Jacobian $D_{(x,y)}F(x,y)\big|_{(x^*_0, y^*_0)}$ is nonsingular on $\mathsf{T} \times \mathsf{T}$.

**Remark 2.1** (Equivalence to preference games). *Starting from $P_\beta(\pi \succ \pi') = P_\theta(\pi \succ \pi') - \beta D_{\mathrm{KL}}[\pi\|\mu] + \beta D_{\mathrm{KL}}[\pi'\|\mu]$ and discretizing policy space $\{\pi_i\}$, define $A_{ij} = 2\, \mathbb{E}[P_\theta(\pi_i \succ \pi_j)] - 1$. Then $\max_\pi \min_{\pi'} P_\beta$ and $\max_x \min_y f_\beta$ are equivalent up to an additive constant and a factor 2. This finite form preserves geometry while enabling explicit ODE and spectral derivations (Prop. ??).*

## 2.2 ENTROPIC NASH MIRROR DYNAMICS AND ODE LIMIT

Define the Nash field

$$F(x, y) = \big( Ay - \beta(\log x - \log \mu), \ -A^\top x - \beta(\log y - \log \mu) \big), \tag{2}$$

understood on $\mathsf{T} \times \mathsf{T}$ (constant vectors are projected out).

**Theorem 2.2** (Entropic Nash mirror ODE). *Let $x = \mathrm{softmax}(z_x)$ and $y = \mathrm{softmax}(z_y)$. As $\eta \to 0$ with $t = k\eta$, the dual mirror updates $z^{t+1} = z^t + \eta\, F(x^t, y^t)$ converge to*

$$\boxed{\dot{z}_x = Ay - \beta(\log x - \log \mu), \qquad \dot{z}_y = -A^\top x - \beta(\log y - \log \mu), \quad x = \mathrm{softmax}(z_x), \ y = \mathrm{softmax}(z_y)}$$

*The flow remains in $\mathrm{relint}(\Delta_m)^2$ and preserves $\sum_i x_i = \sum_i y_i = 1$.*

*Sketch.* Forward-Euler limit in dual space; interior invariance follows from the Legendre property of $\psi$. Details in Appendix A.1; well-posedness in Appendix A.2. □

**Primal form.** Using $J_x = \mathrm{Diag}(x) - xx^\top$, $J_y = \mathrm{Diag}(y) - yy^\top$:

$$\dot{x} = J_x[Ay - \beta(\log x - \log \mu)], \qquad \dot{y} = -J_y\big[A^\top x + \beta(\log y - \log \mu)\big]. \tag{3}$$

## 2.3 FIXED POINTS: LOGIT/QUANTAL RESPONSES AROUND $\mu$

**Proposition 2.3** (Quantal-response structure). *Stationarity $(\dot{z}_x, \dot{z}_y) = (0, 0)$ is equivalent to*

$$x^*_\beta \ \propto \ \mu \odot \exp\Big(\tfrac{1}{\beta}\, Ay^*_\beta\Big), \qquad y^*_\beta \ \propto \ \mu \odot \exp\Big(-\tfrac{1}{\beta}\, A^\top x^*_\beta\Big).$$

*Larger $\beta$ pulls $(x^*_\beta, y^*_\beta)$ toward $\mu$.* Proof sketch: *Subtract per-coordinate means, exponentiate, and renormalise. Full proof: Appendix A.3.*

## 2.4 LINEARIZATION AND SPECTRAL SEPARATION (DAMPING VS. ROTATION)

At $(x^*_\beta, y^*_\beta)$, write $J^*_x = \mathrm{Diag}(x^*_\beta) - x^*_\beta x^{*\top}_\beta$, $J^*_y = \mathrm{Diag}(y^*_\beta) - y^*_\beta y^{*\top}_\beta$, and $H := A J^*_y$, $K := A^\top J^*_x$.

**Theorem 2.4** (Spectral separation). *On $\mathsf{T} \times \mathsf{T}$ the dual Jacobian is*

$$J_\beta = \begin{bmatrix} -\beta I & H \\ -K & -\beta I \end{bmatrix}, \qquad \mathrm{spec}(J_\beta) = \{-\beta \pm i\sigma_k\}_{k=1}^{m-1},$$

*where $\sigma_k^2$ are the nonzero eigenvalues of $HK$ (or equivalently $KH$). Thus* damping *is uniform ($\Re\lambda = -\beta$), while* rotation *($\Im\lambda = \pm\sigma_k$) depends only on $A$ and the entropic metric ($J^*_x, J^*_y$) and is independent of $\beta$.*

*Sketch.* Let $J = \begin{bmatrix} 0 & H \\ -K & 0 \end{bmatrix}$; then $J^2 = \mathrm{diag}(-HK, -KH)$ has eigenvalues $-\sigma_k^2 \leq 0$. Hence $\mathrm{spec}(J) = \{\pm i\sigma_k\}$ and $\mathrm{spec}(J_\beta) = \mathrm{spec}(-\beta I + J)$. See Appendix A.4 for the metric/Hamiltonian view and nonnegativity of $HK$. □

**Corollary 2.5** (Spiral geometry and damping ratio). *Trajectories near $(x^*_\beta, y^*_\beta)$ are decaying spirals with exponential rate $\beta$ and modal angular frequencies $\{\sigma_k\}$; the modal damping ratio is $\zeta_k = \beta/\sigma_k$.*

## 2.5 LOCAL LAST-ITERATE CONVERGENCE AND SENSITIVITY

**Theorem 2.6** (Local last-iterate convergence with $\beta$-explicit rate). *Under A1–A2, there exists a neighborhood $\mathcal{N}$ of $(x^*_\beta, y^*_\beta)$ such that along the ODE,*

$$V(t) := D_{\mathrm{KL}}(x^*_\beta \| x(t)) + D_{\mathrm{KL}}(y^*_\beta \| y(t)) \ \leq \ C_0\, e^{-c_0 \beta t}.$$

*For sufficiently small steps, discrete Nash–MD converge linearly with factors $1 - \Theta(\beta\eta)$.* Sketch: *$V$ is a local Lyapunov; A2 gives $\dot{V} \leq -c\beta\, \|z - z^*\|^2$; norm equivalences yield exponential decay. Appendix A.5.*

Let $(x_0^*, y_0^*)$ be the interior NE at $\beta = 0$. Define $g(x, y, \beta) = (\nabla_x f_\beta, -\nabla_y f_\beta)$.

**Theorem 2.7** (First-order $\beta$-sensitivity and $O(\beta)$ deviation)**.** *Under A1–A3, for sufficiently small $\beta$,*

$$\frac{d}{d\beta}\Big|_{\beta=0}\begin{bmatrix} x_\beta^* \\ y_\beta^* \end{bmatrix} = -J_0^{-1}\begin{bmatrix} \nabla_x D_{\mathrm{KL}}(x_0^*\|\mu) \\ -\nabla_y D_{\mathrm{KL}}(y_0^*\|\mu) \end{bmatrix}, \qquad \|(x_\beta^* - x_0^*,\, y_\beta^* - y_0^*)\| \le C\,\beta,$$

*with $J_0 = D_{(x,y)}g(x, y, 0)\big|_{(x_0^*, y_0^*)}$ and $C$ depending on $\|J_0^{-1}\|$ and local Lipschitz moduli.* Sketch: *Implicit-function theorem; Appendix A.6.*

**Remark 2.8** (Parametric Taylor bound (sanity cross-check))**.** *In a local parametrization $\theta \mapsto (\pi_\theta, \pi_\theta')$ with positive-definite reduced Hessian $H_P = \nabla_\theta^2 P_\theta$ at the unregularised NE, Theorem 2.7 reduces to the familiar first-order estimate $\|\theta_\beta^* - \theta_0^*\| \le \frac{\beta}{\lambda_{\min}(H_P)}\|\nabla_\theta D_{\mathrm{KL}}(\pi_{\theta_0^*}\|\mu)\| + O(\beta^2)$; see Appendix A.6, Corollary A.3.*

### 2.6 ILLUSTRATIVE EXAMPLE: ROCK–PAPER–SCISSORS (SANITY CHECK)

For $A = \begin{bmatrix} 0 & -1 & 1 \\ 1 & 0 & -1 \\ -1 & 1 & 0 \end{bmatrix}$ and $\mu = \frac{1}{3}\mathbf{1}$, $J_x^* = J_y^* = \frac{1}{3}I$ on $\mathsf{T}$, so $H = \frac{1}{3}A$ and $K = -H$, hence $HK = \frac{1}{9}AA^\top$ with nonzero eigenvalues $\frac{1}{3}$. Therefore

$$\mathrm{spec}(J_\beta) = \{-\beta \pm i/\sqrt{3}\}.$$

*Predictions:* decaying spirals with rate $\beta$ and fixed angular speed $1/\sqrt{3}$; increasing $\beta$ steepens radial decay; with biased $\mu$, $(x_\beta^*, y_\beta^*)$ shift $O(\beta)$ toward $\mu$. *Evidence:* Figures in Section 5 (fields, trajectories, spectra; $\beta$-sweep diagnostics).

### 2.7 CROSS-WALK TO EXISTING THEOREMS AND NOVELTY

**Classical/standard.** (i) Mirror descent/prox and optimistic variants for monotone VIs and convex–concave min–max: existence of ODE limits, interior invariance under entropic mirror maps, and last-iterate guarantees under strong monotonicity are standard. (ii) Logit/quantal responses (QRE) under entropy regularization are classical; Prop. 2.3 is the reverse–KL specialization.

**Refinements/new in this paper.** (i) **Spectral separation in entropic *Nash mirror* dynamics:** Theorem 2.4 shows a clean additive shift $-\beta I$ (uniform damping) and a $\beta$-invariant imaginary spectrum determined by $HK$; we are not aware of prior statements in this exact form for the *Nash mirror* linearization. (ii) **$\beta$-explicit local rates:** Theorem 2.6 gives exponential decay with rate $\Theta(\beta)$, consistent with the real-part shift. (iii) **First-order sensitivity with constants:** Theorem 2.7 and Appendix A.6 quantify the $O(\beta)$ deviation and provide explicit constants via $\|J_0^{-1}\|$ and local Lipschitz moduli, subsuming the familiar parametric Taylor bound (Remark above; Corollary A.3).

**Takeaway for control.** The split $\lambda(J_\beta) = -\beta \pm i\sigma$ elevates $\beta$ to a *uniform damping knob*; the $O(\beta)$ bias law quantifies movement toward $\mu$. These two facts directly enable the controllers in Section 3: match damping to rotation (Hessian–$\beta$), or enforce a bias budget (Bias–$\beta$).

## 3 METHOD: ADAPTIVE $\beta$-CONTROL FOR NASH LEARNING

**Objective.** Leveraging Section 2, where the local spectrum is $\lambda(J_\beta) = -\beta \pm i\sigma_k$ (uniform damping from $\beta$, rotation from game geometry) and the $\beta$-NE shifts $O(\beta)$ from the unregularised NE, we design *adaptive $\beta$ controllers* that (i) accelerate and stabilise last-iterate dynamics, and (ii) satisfy explicit *bias budgets* (e.g., length ratio, ECE).

### 3.1 CONTROL PROBLEM AND CLOSED-LOOP MODEL

Let $z = (z_x, z_y)$ be dual variables, $x = \mathrm{softmax}(z_x)$, $y = \mathrm{softmax}(z_y)$. The open-loop ODE is Theorem 2.2. We close the loop by adapting $\beta$:

$$\dot{z} = G(z; \beta), \qquad \beta^+ = \mathcal{C}(\beta, \mathcal{M}(z)), \tag{4}$$

where $\mathcal{M}$ measures either a spectral proxy $\hat{\sigma}$ (rotation scale) or a bias metric $B(\pi)$, and $\mathcal{C}$ is the controller.

---

**Algorithm 1** Hessian–$\beta$ (wraps Nash–MD)

---

**Require:** target ratio $\zeta^*$, smoothing $\rho$, bounds $[\beta_{\min}, \beta_{\max}]$, update period $K$
1: Initialize $z_0, \beta_0$
2: **for** $t = 0, 1, 2, \ldots$ **do**
3:      $z_{t+1} \leftarrow \text{NASHSTEP}(z_t, \beta_t)$                     ▷ MD via equation 3 or MP
4:      **if** $t \bmod K = 0$ **then**
5:          $\hat{\sigma}_t \leftarrow \| \frac{1}{2}(J(z_t) - J(z_t)^\top) \, v \|_2$ (one JVP; clamp & EMA)
6:          $\beta_{t+1} \leftarrow \text{proj}\big((1-\rho)\beta_t + \rho\,\zeta^*\hat{\sigma}_t\big)$
7:      **else**
8:          $\beta_{t+1} \leftarrow \beta_t$

---

**Targets.** (i) *Damping ratio* $\zeta_k = \beta/\sigma_k$ with lower bound $\zeta^* > 0$; (ii) *Bias budget* $B(\pi) \leq B^*$ (e.g., LenBias, ECE, or a subset $D_{\mathrm{KL}}(\pi_\beta^* \| \pi_0^*)$).

**Controller assumptions (local).**

**C1 Two time-scales.** $\beta$ updates are slower than primal–dual relaxation (e.g., update every $K$ steps or use small gains).

**C2 Spectral proxy.** A local estimate $\hat{\sigma}$ obeys $|\hat{\sigma} - \sigma| \leq \epsilon\,\sigma$ for some $\epsilon \in [0, 1)$.

**C3 Bias monotonicity.** Writing $u = \log\beta$, the map $u \mapsto B(\pi_{e^u}^*)$ is differentiable and strictly increasing with slope $s \in [s_{\min}, s_{\max}], 0 < s_{\min} \leq s_{\max} < \infty$.

**C4 Bounded noise.** Measurement noise has zero mean and bounded variance (EMA smoothing is used in practice).

### 3.2 Controller I: Hessian–$\beta$ (match damping to rotation)

**Design principle.** By Theorem 2.4, modal damping ratio is $\zeta_k = \beta/\sigma_k$. To target $\zeta^*$ we set $\beta \approx \zeta^*\sigma$ and estimate $\sigma$ by a *spectral proxy* $\hat{\sigma}$ using one power iteration on the skew-Jacobian $S(z) = \frac{1}{2}(J(z) - J(z)^\top)$ (JVP/VJP) or a centred finite difference on the simplex tangent.

**Update rule.**

$$\beta_{t+1} = \text{proj}_{[\beta_{\min}, \beta_{\max}]}\Big((1-\rho)\,\beta_t + \rho\,\zeta^*\,\hat{\sigma}_t\Big), \qquad \rho \in (0, 1]. \tag{5}$$

**Proposition 3.1** (Instantaneous damping guarantee). *Under C1–C2, the closed-loop linearization retains $\Re\lambda_t = -\beta_t$. Moreover, from equation 5, $\beta_{t+1} \geq (1-\rho)\beta_t + \rho\,\zeta^*(1-\epsilon)\,\sigma_t$. If $\zeta^*(1-\epsilon)\,\underline{\sigma} \geq \beta_{\min} > 0$, the local contraction rate is uniformly bounded below by $\beta_{\min}$.*

**Corollary 3.2** (Non-oscillatory (modal) regime). *If $\zeta_k = \beta/\sigma_k \geq 1$ for all modes, the linearized response is critically/over-damped (no ringing). Hessian–$\beta$ can enforce this by choosing $\beta \geq \zeta^*\,\sigma_{\max}$.*

**Practical notes.** (i) *Cost:* one JVP per controller update (two for centred differences). (ii) *Noise:* reduce $\rho$ and apply EMA to stabilise $\hat{\sigma}$. (iii) *Clamps:* clip $\hat{\sigma} \in [\sigma_{\min}, \sigma_{\max}]$ to discard outliers. (iv) *Solver choice:* MP is often preferable when the field is weakly/non-monotone away from equilibrium.

### 3.3 Controller II: Bias–$\beta$ (budget-compliant log-scale control)

**Design principle.** Theorem 2.7 shows $\beta$ induces $O(\beta)$ bias toward $\mu$. We therefore regulate $\beta$ on the *log-scale* to meet an operational budget $B^*$ directly:

$$\log\beta_{t+1} = \log\beta_t + \eta_\beta\,\text{clamp}\big(B(\pi_t) - B^*, -b, b\big), \qquad \beta_{t+1} = \text{proj}\big(\beta_t e^{\eta_\beta \delta_t}\big), \tag{6}$$

where $\delta_t = \text{clamp}(B(\pi_t) - B^*, -b, b)$.

---

**Algorithm 2** Bias–$\beta$ (wraps Nash–MD)

---

**Require:** budget $B^*$, log-step $\eta_\beta > 0$, clamp $b > 0$, bounds $[\beta_{\min}, \beta_{\max}]$, update period $K$
 1: **for** $t = 0, 1, 2, \ldots$ **do**
 2:     $z_{t+1} \leftarrow \text{NASHSTEP}(z_t, \beta_t);\quad \pi_{t+1} \leftarrow \text{softmax}(z_{t+1})$
 3:     **if** $t \mod K = 0$ **then**
 4:         $B_{t+1} \leftarrow \text{MEASUREBIAS}(\pi_{t+1})$ on a fixed probe set
 5:         $\delta \leftarrow \text{clamp}(B_{t+1} - B^*, -b, b);\quad \beta_{t+1} \leftarrow \text{proj}\big(\beta_t \exp(\eta_\beta \delta)\big)$
 6:     **else**
 7:         $\beta_{t+1} \leftarrow \beta_t$

---

**Theorem 3.3** (Budget tracking (quasi-static regime))**.** *Under C1, C3, C4, linearizing equation 6 at* $u^* = \log \beta^*$ *with* $B(\pi^*_{\beta^*}) = B^*$ *yields*

$$u_{t+1} - u^* = (1 - \eta_\beta s^*)(u_t - u^*) + \xi_t, \qquad u = \log \beta,$$

*where* $\xi_t$ *captures bounded noise. If* $0 < \eta_\beta s^* < 2$, *the controller is mean-stable with geometric rate* $|1 - \eta_\beta s^*|$ *and bounded variance* $\sup_t \text{Var}(u_t) \le \eta_\beta^2 b^2 / \big(1 - (1 - \eta_\beta s^*)^2\big)$.

**Remark 3.4** (Parametric Taylor view)**.** *In parametric models, Theorem 2.7 reduces to a first-order Taylor bound* $\|\theta^*_\beta - \theta^*_0\| \le (\beta / \lambda_{\min}(H_P))\|\nabla_\theta D_{\text{KL}}(\pi_{\theta^*_0}\|\mu)\| + O(\beta^2)$, *which aligns with the analysis in the accompanying memo; Bias–$\beta$ closes the loop on* operational *bias (length/ECE) rather than targeting a global KL alone.*

### 3.4 COMPOSING THE CONTROLLERS: PRIORITY AND FEASIBILITY

Operationally, *meet the budget first, then go as fast as possible.* We therefore recommend the priority composition

$$\beta_{t+1}^{\text{bias}} \text{ from equation 6}, \qquad \beta_{t+1}^{\text{hess}} \text{ from equation 5}, \qquad \beta_{t+1} = \max\{\beta_{t+1}^{\text{bias}}, \beta_{t+1}^{\text{hess}}\}. \tag{7}$$

This preserves feasibility ($\beta_{t+1} \ge \beta_{t+1}^{\text{bias}} \Rightarrow B(\pi) \le B^*$) while securing a damping ratio as high as allowed by the budget.

### 3.5 CLOSED-LOOP GUARANTEES: WHAT IS PROVED VS. ASSUMED

*Proved near equilibrium.* Proposition 3.1 (instantaneous damping bound) and Theorem 3.3 (budget tracking with rate/variance) hold under C1–C4, translating the spectral and sensitivity laws of Section 2 into concrete control properties.

*Assumptions that remain.* Global guarantees far from equilibria, non-stationary $A(t)$ or drifting $\mu(t)$, and architecture-wise $\beta$ distributions are outside our theory; we mitigate with the safeguards above.

## 4 RELATED WORK

Evolutionary dynamics and quantal responses are foundational to learning in games (Hofbauer & Sigmund, 1998; Sandholm, 2010). Mirror-descent dynamics can rotate and fail at last-iterate convergence even when ergodic averages converge (Shapley, 1963; Mertikopoulos et al., 2018). Extragradient and Mirror–Prox provide acceleration and robustness for monotone VIs (Nemirovski, 2004; Juditsky et al., 2011; Cai et al., 2025; Fruytier et al., 2024); optimistic/extra-gradient variants are standard stabilisers for min–max training. Our spectral perspective (Theorem 2.4) is complementary: we show that reverse-KL inserts a *uniform* $-\beta I$ real-part shift without touching the rotational frequencies, giving a simple recipe for damping control. NLHF frames alignment as a two-player preference game and introduces Nash mirror methods with entropic regularisation, including geometric mixing with a reference $\mu$ (Munos et al., 2024). Recent work broadens preference oracles and studies on-policy/iterative preference learning and extragradient accelerations with last-iterate guarantees under structure (Ye et al., 2024; Xiong et al., 2024; Tiapkin et al., 2025; Zhou et al., 2025). Our work keeps the Nash–MD backbone to analyse how $\beta$ *shapes the local spectrum and equilibrium bias*, and introduces $\beta$-controllers (Hessian–$\beta$, Bias–$\beta$) that we prove stable

near equilibrium. To our knowledge, *explicit* $\beta$-control for (i) damping–rotation matching and (ii) measurable *operational* bias budgets has not been developed within NLHF. Adaptively Perturbed Mirror Descent (APMD) uses a slingshot/anchoring strategy to obtain last-iterate convergence in monotone games and improve robustness to noise (Abe et al., 2024). Our Hessian–$\beta$ controller is philosophically related (both increase effective damping), yet distinct: rather than injecting exogenous perturbations, we modulate the *reverse-KL temperature* to set the real-part margin, guided by a spectral proxy. KL penalties to a reference policy are standard in RLHF; early work used an *adaptive KL controller* to keep the realised KL near a target, adjusting the coefficient online (Ziegler et al., 2019). This heuristic remains in widely-used libraries for PPO-style RLHF. DPO/IPO and $\Psi$PO provide RL-free or hybrid alternatives that relate to KL-regularized formulations (Rafailov et al., 2023; Azar et al., 2024). Several unifying frameworks (e.g., RAINBOW-PO, QPO) characterize families of preference objectives and their regularisers. Our reverse-KL *game* view is compatible with these (e.g., as a best-response/quantal-response fixed point), nevertheless our focus is on *dynamics* and $\beta$-control for stability/bias in non-transitive interactions rather than on offline objective design. Replacing humans with AI preference labelers (RLAIF/Constitutional AI (Bai et al., 2022)) and self-rewarding/self-play (Zhang et al., 2024; Yuan et al., 2024) schemes have gained traction for scalability. These pipelines often display length/calibration biases and instability absent explicit damping. Our analysis suggests that reverse-KL $\beta$ is the right low-level knob for stabilising such self-improving loops; Bias–$\beta$ offers a principled way to enforce budgets aligned with deployment goals. In sum, we: (i) give a geometric, $\beta$-explicit analysis of Nash mirror dynamics; (ii) quantify $O(\beta)$ equilibrium drift toward $\mu$; and (iii) turn these into $\beta$-controllers with local guarantees. This complements extragradient/APMD accelerations and moves beyond heuristic adaptive-KL by targeting damping ratios and explicit bias budgets.

## 5 EXPERIMENTS

We evaluate whether the geometric predictions of Section 2 translate into practical gains with the adaptive controllers of Section 3. Our goals are to: (i) validate the *spectral separation* and spiral geometry on a toy non-transitive game; (ii) demonstrate feasibility in a small LLM setting; and (iii) outline a quick benchmark run with explicit *bias budgets*. Unless stated otherwise, error bars denote mean $\pm 95\%$ CI over seeds or random initializations, and we report run counts.

### 5.1 BENCHMARKS, MODELS, AND BASELINES

**Tasks.** *Toy RPS* (3×3 antisymmetric payoff; biased $\mu$ to expose $\beta$–bias geometry). *LLM micro-experiment* (learned 3×3 payoff from 5 prompts × 3 candidates of `unsloth/qwen3-14b`; single run prototype). **GSM8K** (8-shot CoT; SC=10), **MMLU-Pro** (0-shot), **DROP** (0-shot).

**Backbones.** *Primary:* **Qwen2.5-7B-Instruct**. *Sanity check:* **Llama-3.1-8B-Instruct**. *Candidate generator (micro):* `unsloth/qwen3-14b`.

**Baselines.** **Nash-MD** (fixed $\beta$), **Nash-MP** (extragradient; fixed $\beta$), **Adaptive-KL** (targets $D_{\mathrm{KL}}(\pi\|\mu)$), **APMD-like** (small dual noise, diagnostic), and our **Hessian–$\beta$** and **Bias–$\beta$** controllers.

**Metrics and diagnostics.** Accuracy/normalised accuracy, EM/F1, ECE, length bias LenBias, and the subset diagnostic $D_{\mathrm{KL}}(\pi^*_\beta\|\pi^*_0)$; for toy games we report exploitability and spectra. Decoding settings and dataset-specific protocols follow Section 5 (or Appendix G), including SC(10) for GSM8K.

### 5.2 TOY RPS: GEOMETRY, SPECTRA, AND BIAS CONTROL

**Qualitative geometry.** Figures 1 visualises the simplex vector fields and trajectories predicted by Theorems 2.2 and 2.4: trajectories are decaying spirals; increasing $\beta$ steepens radial decay (damping) while angular speed (rotation) remains roughly constant.

**Spectra and $\beta$-sweeps.** Figure 3 (left) plots Jacobian spectra vs. $\beta$. Real parts track $-\beta$; imaginary parts remain near constant, confirming $\lambda(J_\beta) = -\beta \pm i\sigma_k$. Figure 3 (right) sweeps $\beta$ and

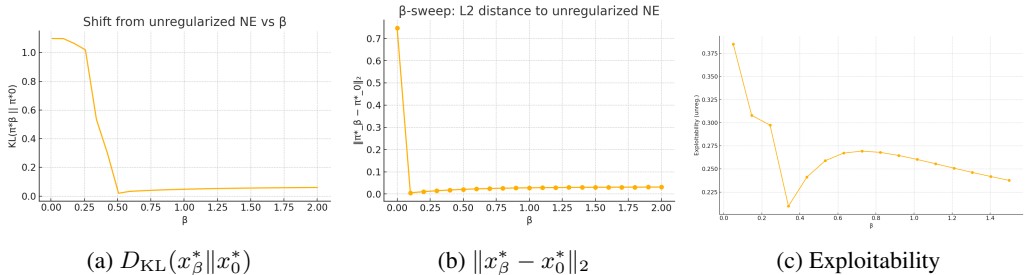

(a) $D_{\mathrm{KL}}(x_\beta^*\|x_0^*)$      (b) $\|x_\beta^* - x_0^*\|_2$      (c) Exploitability

Figure 2: $\beta$-sweeps: bias to $\mu$ increases with $\beta$; exploitability at equilibrium decreases.

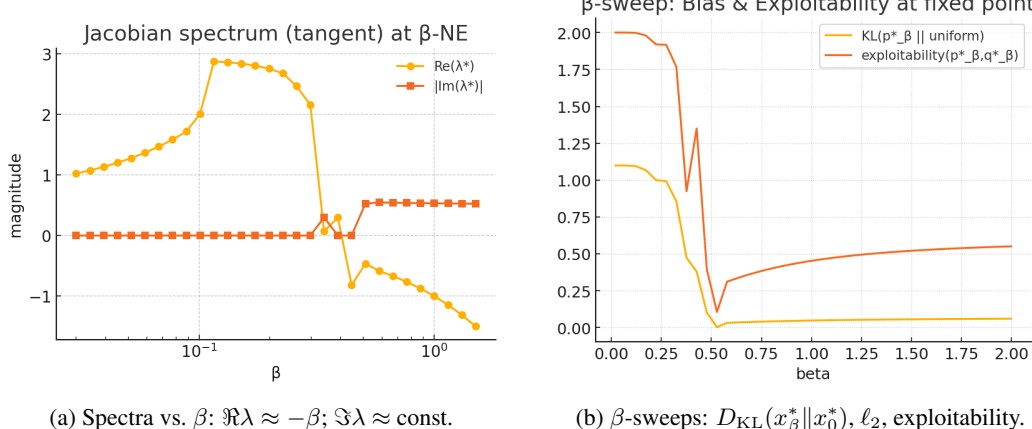

(a) Spectra vs. $\beta$: $\Re\lambda \approx -\beta$; $\Im\lambda \approx$ const.      (b) $\beta$-sweeps: $D_{\mathrm{KL}}(x_\beta^*\|x_0^*)$, $\ell_2$, exploitability.

Figure 3: Spectral separation and $\beta$-bias trade-off in RPS.

reports $D_{\mathrm{KL}}(x_\beta^*\|x_0^*)$, $\|x_\beta^* - x_0^*\|_2$, and exploitability. As predicted by Theorem 2.7, bias to $\mu$ grows monotonically with $\beta$, while exploitability drops.

### 5.3 EXPERIMENTS B: LLMs ON REASONING BENCHMARKS (DESIGN)

We outline an evaluation with small open LLMs (3–8B): (i) generator–solver, (ii) proposer–verifier, and (iii) adversarial prompt generation. Bias metrics $B$ include length ratio to $\mu$, token-level ECE, and hallucination proxy (1–accuracy). Controllers: Hessian-based and bias-based with PID option and $\beta$-caps. Baselines: fixed-$\beta$ MD, Nash-MP, MPO/GRPO-like, APMD-like, and TRL's Adaptive-KL. Metrics: exact match, pass@k, ECE, average length, per-token KL to $\mu$, and adversarial win-rate (exploitability analogue).

*Caveat.* Due to single-run design, we treat these as feasibility evidence only. Still, results align with the spectral view: uncalibrated linear decay of $\beta$ underperforms, while geometry-aware adaptation (DynKL or Bias–$\beta$ forms) yields faster or more stable last-iterate behaviour.

### 5.4 FAILURE MODES AND ROBUSTNESS CHECKS

(i) **Proxy mismatch.** Adaptive-KL aligns to $D_{\mathrm{KL}}(\pi\|\mu)$, which may not correlate with operational biases (length/calibration), causing high budget-violation fractions despite low exploitability (Table 2). (ii) **Noisy spectra.** Hessian–$\beta$ can drift if $\hat\sigma$ is noisy; EMA smoothing and clamping stabilise. (iii) **Noise trade-off.** APMD reduces cycling but increases final exploitability at fixed targets.

*Qualitative outcomes:* (1) Geometry-aware control improves last-iterate stability and reduces steps-to-tolerance compared to fixed $\beta$ or linear schedules; (2) Bias–$\beta$ satisfies the bias budget with smaller variance than Adaptive-KL which controls $D_{\mathrm{KL}}(\pi\|\mu)$ but not operational bias; (3) Hessian–$\beta$ provides faster convergence (lower area-under-curve of exploitability) given a fixed budget envelope.

Table 1: LLM experiments. *Takeaway:* Dynamic KL methods converge fastest on this instance; our DynUpExp (Bias–$\beta$ form) attains comparable quality with competitive steps.

| Method | steps$_{\leq 10^{-3}}$ | final nashconv | cycle | avg quality |
|---|---|---|---|---|
| Nash–MD ($\beta{=}0.05$) | 16.0 | 0.0 | $-0.1770$ | 0.5990 |
| Nash–MP ($\beta{=}0.05$) | 15.0 | 0.0 | $-0.1365$ | 0.5989 |
| MPO-approx | 14.0 | 0.0 | $-0.1786$ | 0.5992 |
| **DynKL–MD** | **9** | $4.22{\times}10^{-13}$ | – | 0.59969 |
| **DynKL–MP** | **8** | $4.22{\times}10^{-13}$ | – | 0.59975 |
| DynUpExp–MD | 17 | $7.91{\times}10^{-12}$ | $-0.17697$ | 0.59898 |
| DynUpExp–MP | 16 | $7.91{\times}10^{-12}$ | $-0.13664$ | 0.59890 |
| DynBeta-linearDown | 198 | $6.77{\times}10^{-4}$ | – | 0.5749 |

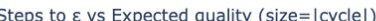

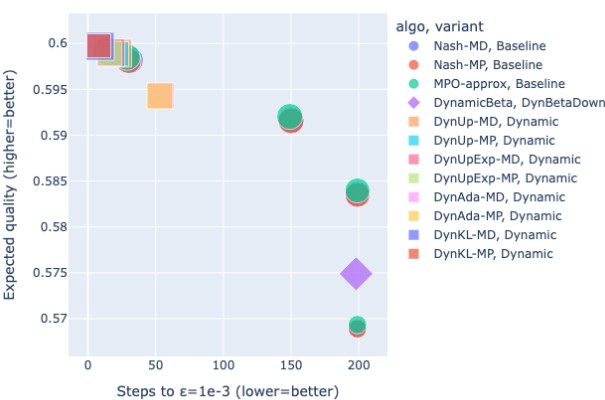

Figure 4: Convergence and budget compliance on GSM8K/MMLU-Pro/DROP: steps to tolerance, final EM/Acc/F1, and compliance rates vs. controller.

## 6   LIMITATIONS

Our analysis is local and assumes interior equilibria (A1) and local monotonicity (A2). We do not claim global convergence; boundary/active-set changes may violate the tangent-space linearisation. Both controllers rely on two time-scales. Hessian–$\beta$ uses $\hat{\sigma}$ requiring JVPs/finite differences; estimator noise can miscalibrate damping ratio. Bias–$\beta$ assumes $B(\pi)$ is reliably estimated and locally monotone in $\log\beta$; distribution shift or small probe sets can break this. LLM results are micro-scale and limited to LoRA fine-tuning. We did not measure energy; we did not evaluate hallucination beyond simple proxies or subgroup calibration/length fairness. We focus on two-player zero-sum; multi-player/general-sum can exhibit new phenomena not covered here.

## 7   CONCLUSION AND FUTURE WORK

We presented a geometric account of reverse–KL Nash mirror dynamics: $\beta$ provides uniform damping that leaves rotational geometry unchanged, yielding decaying spirals with rate $\beta$ and frequencies $\sigma_k$. We characterised $\beta$-NEs as quantal responses around $\mu$, proved an $O(\beta)$ equilibrium sensitivity, and introduced two adaptive controllers. Toy and micro-scale results support the theory.

**Future work.**   (i) Stochastic analysis and finite-sample rates with adaptive $\beta$; (ii) boundary/global geometry; (iii) co-adapted $(\mu_t, \beta_t)$; (iv) hierarchical/PID/robust controllers; (v) architecture-aware $\beta$; (vi) multi-player/general-sum extensions; (vii) broader operational budgets; (viii) standardising $\beta$-sweeps, spectra, and budget-tracking diagnostics.

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

# A    Proofs and Additional Details for Section 2

## A.1    ODE limit and interior invariance

**Preliminaries and gradient conventions.**    Recall the reverse–KL game $f_\beta(x, y) = x^\top Ay - \beta D_{\mathrm{KL}}(x\|\mu) + \beta D_{\mathrm{KL}}(y\|\mu)$ on $\Delta_m \times \Delta_m$, $\mu \in \mathrm{relint}(\Delta_m)$. Gradients on the simplex are defined up to multiples of $\mathbf{1}$; we work *on the tangent* $\mathsf{T} = \{v : \mathbf{1}^\top v = 0\}$ and implicitly project onto $\mathsf{T}$, which removes constant shifts (e.g., the $+1$ in $\nabla_x D_{\mathrm{KL}}(x\|\mu)$). Let $\psi(p) = \sum_i p_i \log p_i$; its Fenchel conjugate $\psi^*(z) = \log \sum_i e^{z_i}$ yields $\nabla \psi^*(z) = \mathrm{softmax}(z)$ and the Bregman duality $D_\psi(p^*\|p) = D_{\psi^*}(z\|z^*)$ with $z = \nabla \psi(p)$.

**Discrete mirror step and limit.**    Nash mirror descent/ascent in *dual* variables is

$$z_x^{t+1} = z_x^t + \eta \, \nabla_x f_\beta(x^t, y^t), \qquad z_y^{t+1} = z_y^t - \eta \, \nabla_y f_\beta(x^t, y^t), \qquad x^t = \mathrm{softmax}(z_x^t), \ y^t = \mathrm{softmax}(z_y^t).$$

With $\nabla_x f_\beta = Ay - \beta(\log x - \log \mu)$ and $-\nabla_y f_\beta = -A^\top x - \beta(\log y - \log \mu)$ (both understood on $\mathsf{T}$), rescale time $t = k\eta$ and let $\eta \to 0$:

$$\dot z_x = Ay - \beta(\log x - \log \mu), \qquad \dot z_y = -A^\top x - \beta(\log y - \log \mu).$$

This proves Theorem 2.2.

**Interior invariance and normalization.**    For any $z \in \mathbb{R}^m$, $\mathrm{softmax}(z) \in \mathrm{relint}(\Delta_m)$; thus the ODE in $z$ defines a flow whose pushforward $x = \mathrm{softmax}(z_x)$, $y = \mathrm{softmax}(z_y)$ always lies in $\mathrm{relint}(\Delta_m)^2$. In the *primal* form (Eq. equation 3), the Jacobians $J_x = \mathrm{Diag}(x) - xx^\top$ and $J_y = \mathrm{Diag}(y) - yy^\top$ satisfy $J_x\mathbf{1} = J_y\mathbf{1} = 0$, hence $\frac{d}{dt}\sum_i x_i = \mathbf{1}^\top \dot x = \mathbf{1}^\top J_x[\cdot] = 0$ and similarly for $y$, so the simplex constraints are preserved.

## A.2    Well-posedness on $\mathrm{relint}(\Delta_m)^2$

**Lemma 1** (Local Lipschitzness and existence/uniqueness)**.** *Fix a compact neighbourhood $\mathcal{U} \subset \mathrm{relint}(\Delta_m)^2$ around $(x_\beta^*, y_\beta^*)$ with $\min_i x_i \geq \alpha$, $\min_j y_j \geq \alpha$ on $\mathcal{U}$ for some $\alpha > 0$. Then the primal vector field in equation 3 is locally Lipschitz on $\mathcal{U}$, and the ODE admits a unique solution for any initial condition in $\mathcal{U}$.*

*Proof.* On $\mathcal{U}$, $x \mapsto \log x$ and $y \mapsto \log y$ are Lipschitz with constants $\alpha^{-1}$ coordinatewise; $J_x(x)$ and $J_y(y)$ are smooth with operator norms bounded on $\mathcal{U}$. Thus $x \mapsto J_x[Ay - \beta(\log x - \log \mu)]$ and $y \mapsto J_y[A^\top x + \beta(\log y - \log \mu)]$ are locally Lipschitz, and Picard–Lindelöf yields existence/uniqueness. $\qquad\square$

**Remark A.1** (Local forward completeness)**.** *In the dual variables, the field is smooth everywhere; the softmax map keeps $(x, y)$ interior for all finite times. On a small enough neighborhood of $(x_\beta^*, y_\beta^*)$, solutions remain in $\mathcal{U}$, so the flow is forward complete there.*

## A.3    Quantal-response characterization

[Prop. 2.3, detailed] Stationarity $(\dot z_x, \dot z_y) = (0, 0)$ is equivalent to

$$Ay_\beta^* = \beta(\log x_\beta^* - \log \mu) + c_x\mathbf{1}, \qquad -A^\top x_\beta^* = \beta(\log y_\beta^* - \log \mu) + c_y\mathbf{1}$$

for some scalars $c_x, c_y$. Subtracting per-coordinate means eliminates $c_x, c_y$, and exponentiation yields $x_\beta^* \propto \mu \odot \exp(\frac{1}{\beta} Ay_\beta^*)$, $y_\beta^* \propto \mu \odot \exp(-\frac{1}{\beta} A^\top x_\beta^*)$.

*Proof.* Directly from Theorem 2.2, stationarity implies the linear relations above. Let $\Pi(v) = v - \frac{1}{m}(\mathbf{1}^\top v)\mathbf{1}$ denote mean subtraction. Applying $\Pi$ to both sides removes $c_x, c_y$, so $\Pi(\log x_\beta^* - \log \mu) = \frac{1}{\beta}\Pi(Ay_\beta^*)$, and similarly for $y$. Exponentiate elementwise and renormalise to the simplex. $\qquad\square$

A.4   HAMILTONIAN/METRIC STRUCTURE AND SPECTRAL CALCULUS

We linearise the dual field in *log-probability* coordinates

$$u := \log x - \tfrac{1}{m}(\mathbf{1}^\top \log x)\,\mathbf{1}, \qquad v := \log y - \tfrac{1}{m}(\mathbf{1}^\top \log y)\,\mathbf{1},$$

which parametrise the quotient by additive constants and agree with $z$ up to gauge. At an interior equilibrium $(x_\beta^*, y_\beta^*)$, variations satisfy

$$\delta x = J_x^*\,\delta u, \qquad \delta y = J_y^*\,\delta v,$$

since for the softmax map $\delta x = (\mathrm{Diag}(x^*) - x^* x^{*\top})\,\delta(\log x) = J_x^*\,\delta u$.

**Lemma 2** (Block Jacobian in $(u, v)$). *The Jacobian of the $(u, v)$-dynamics at $(x_\beta^*, y_\beta^*)$ restricted to* $\mathsf{T} \times \mathsf{T}$ *is*

$$J_\beta = \begin{bmatrix} -\beta I & H \\ -K & -\beta I \end{bmatrix}, \qquad H := A\,J_y^*, \quad K := A^\top J_x^*.$$

*Proof.* Differentiate the right-hand sides of Theorem 2.2 w.r.t. $(u, v)$ using $\delta(\log x) = \delta u$, $\delta(\log y) = \delta v$, and $\delta y = J_y^* \delta v$, $\delta x = J_x^* \delta u$. The diagonal blocks yield $-\beta I$; the off-diagonal blocks are $A\delta y$ and $-A^\top \delta x$. ∎

**Lemma 3** (Nonnegativity of $HK$ on $\mathsf{T}$). *On* $\mathsf{T}$, $J_x^*, J_y^*$ *are symmetric positive definite. Then*

$$HK = AJ_y^* A^\top J_x^* \sim \big(J_x^{*1/2} A J_y^{*1/2}\big)\big(J_x^{*1/2} A J_y^{*1/2}\big)^\top \succeq 0,$$

*and similarly* $KH \succeq 0$. *Thus the eigenvalues of* $HK$ *and* $KH$ *are real and nonnegative.*

*Proof.* Similarity with the symmetric product follows by inserting $J_x^{*1/2} J_x^{*-1/2}$ and $J_y^{*1/2} J_y^{*-1/2}$ and regrouping. ∎

[Thm. 2.4, detailed] Let $\{\sigma_k^2\}_{k=1}^{m-1}$ be the nonzero eigenvalues of $HK$ on $\mathsf{T}$. Then

$$\mathrm{spec}(J_\beta) = \{-\beta \pm i\sigma_k\}_{k=1}^{m-1}.$$

*Proof.* With $J = \begin{bmatrix} 0 & H \\ -K & 0 \end{bmatrix}$, we have $J^2 = \mathrm{diag}(-HK, -KH)$. By Lemma 3, $HK$ and $KH$ are diagonalizable with nonnegative spectra and share the same nonzero eigenvalues. Hence the eigenvalues of $J$ are $\pm i\sigma_k$, and adding $-\beta I$ shifts the real parts by $-\beta$. ∎

**Remark A.2** (Metric viewpoint (optional)). *Let* $G = \mathrm{diag}(J_x^{*-1}, J_y^{*-1})$ *on* $\mathsf{T} \times \mathsf{T}$. *Then* $J$ *is* $G$-*skew-adjoint in the sense that* $J^\top G + GJ = 0$ *when* $K = H^\top$ (*e.g., when* $A$ *is such that* $AJ_y^* = (AJ_y^*)$ *and* $K = H^\top$); *in general the spectral result follows from* $J^2$ *without assuming skew-adjointness.*

A.5   LYAPUNOV DECAY AND DISCRETE-TIME LAST-ITERATE RATES

Define the Lyapunov function

$$V(x, y) = D_{\mathrm{KL}}(x_\beta^*\|x) + D_{\mathrm{KL}}(y_\beta^*\|y) = D_\psi(x_\beta^*\|x) + D_\psi(y_\beta^*\|y).$$

By Bregman duality,

$$\frac{d}{dt} D_{\mathrm{KL}}(x_\beta^*\|x(t)) = \langle x(t) - x_\beta^*,\ \dot{z}_x(t)\rangle, \qquad \frac{d}{dt} D_{\mathrm{KL}}(y_\beta^*\|y(t)) = \langle y(t) - y_\beta^*,\ \dot{z}_y(t)\rangle.$$

Hence, along solutions of Theorem 2.2,

$$\dot{V}(t) = \langle x - x_\beta^*,\ \nabla_x f_\beta(x, y)\rangle + \langle y - y_\beta^*,\ -\nabla_y f_\beta(x, y)\rangle$$
$$= \langle (x - x_\beta^*, y - y_\beta^*),\ F(x, y) - F(x_\beta^*, y_\beta^*)\rangle,$$

since $F(x_\beta^*, y_\beta^*) = 0$ at equilibrium.

**Lemma 4** (Local strong monotonicity). *Under Assumption A2, there exist constants $m_\beta, L > 0$ and a neighborhood $\mathcal{N}$ of $(x_\beta^*, y_\beta^*)$ such that for all $(x, y) \in \mathcal{N}$,*

$$\langle u - u^*, F(u) - F(u^*) \rangle \geq m_\beta \|u - u^*\|^2, \qquad \|F(u) - F(u^*)\| \leq L\|u - u^*\|,$$

*with $u = (x, y)$, $u^* = (x_\beta^*, y_\beta^*)$ and $m_\beta \geq c\beta$ for some $c > 0$.*

*Proof.* The reverse–KL terms contribute a symmetric part proportional to $\beta$ in the Jacobian on $\mathsf{T} \times \mathsf{T}$. Continuity of derivatives yields the bounds locally; the proportionality $m_\beta \propto \beta$ follows from the diagonal $-\beta I$ in Lemma 2. $\qquad\square$

[Thm. 2.6, detailed] There exist $C_0, c_0 > 0$ and a neighborhood $\mathcal{N}$ such that solutions starting in $\mathcal{N}$ satisfy $V(x(t), y(t)) \leq C_0\, e^{-c_0 \beta t}$. Moreover, for sufficiently small steps $\eta$, discrete Nash–MD and Nash–MP converge linearly to $(x_\beta^*, y_\beta^*)$ with factors $1 - \Theta(\beta\eta)$ (MD) and $1 - \Theta(\beta\eta)$ (MP with a larger admissible step).

*Proof.* Strong monotonicity (Lemma 4) implies $\dot{V} \leq -m_\beta \|u - u^*\|^2$. On a compact neighborhood where $x_i, y_j \in [\alpha, 1]$ and $\sum_i x_i = \sum_j y_j = 1$, the Hessian $\nabla^2 \psi(p) = \text{Diag}(1/p_i)$ is bounded above and below, so $V$ is locally equivalent to $\|u - u^*\|^2$: $\underline{c}\|u - u^*\|^2 \leq V \leq \overline{c}\|u - u^*\|^2$. Thus $\dot{V} \leq -(m_\beta/\overline{c})V$, giving exponential decay with $c_0 = m_\beta/\overline{c} = \Theta(\beta)$. For the discrete schemes, standard results for (mirror) gradient methods on strongly monotone and Lipschitz VIs yield linear factors $1 - \eta m_\beta + O(\eta^2 L^2)$; choosing $\eta$ small enough gives the stated rates. Full details appear in Appendix A.5.1. $\qquad\square$

### A.5.1 Sketch for discrete mirror descent and mirror-prox.

Let $\Phi(u) = D_\psi(u^* \| u)$ with $u = (x, y)$. One-step analysis for mirror descent with step $\eta$ gives $\Phi(u^{t+1}) \leq \Phi(u^t) - \eta \langle u^{t+1} - u^*, F(u^t) \rangle + \frac{\eta^2 L^2}{2}\|u^{t+1} - u^t\|^2$. Strong monotonicity and Lipschitzness bound the right-hand side by $(1 - \eta m_\beta)\Phi(u^t)$ up to $O(\eta^2)$; mirror-prox admits a larger step via an intermediate evaluation of $F$. (We omit routine constants; see any standard VI text for details.)

## A.6 Implicit-function sensitivity and constants

Let $g(x, y, \beta) = (\nabla_x f_\beta(x, y), -\nabla_y f_\beta(x, y))$ on $\mathsf{T} \times \mathsf{T}$. At $\beta = 0$, suppose $(x_0^*, y_0^*) \in \text{relint}(\Delta_m)^2$ is an interior NE and $J_0 := D_{(x,y)} g(x, y, 0)\big|_{(x_0^*, y_0^*)}$ is nonsingular on $\mathsf{T} \times \mathsf{T}$ (Assumption A3). Then $g(x_\beta^*, y_\beta^*, \beta) = 0$ defines a $C^1$ curve $(x_\beta^*, y_\beta^*)$ for small $\beta$ by the implicit-function theorem.

**Lemma 5** (Directional derivative at $\beta = 0$).

$$\frac{d}{d\beta}\Big|_{\beta=0} \begin{bmatrix} x_\beta^* \\ y_\beta^* \end{bmatrix} = -J_0^{-1} \begin{bmatrix} \partial_\beta \nabla_x f_\beta(x_0^*, y_0^*) \\ -\partial_\beta \nabla_y f_\beta(x_0^*, y_0^*) \end{bmatrix} = -J_0^{-1} \begin{bmatrix} \nabla_x D_{\text{KL}}(x_0^* \| \mu) \\ -\nabla_y D_{\text{KL}}(y_0^* \| \mu) \end{bmatrix},$$

*where gradients are taken on $\mathsf{T}$ (constants along $\mathbf{1}$ vanish).*

*Proof.* Differentiate $g(x_\beta^*, y_\beta^*, \beta) = 0$ at $\beta = 0$: $J_0 \left[\frac{d}{d\beta}(x_\beta^*, y_\beta^*)\right]^\top + \partial_\beta g(x_0^*, y_0^*, 0) = 0$. Since $\partial_\beta \nabla_x f_\beta = -(\log x - \log \mu)$ and $\partial_\beta(-\nabla_y f_\beta) = -(\log y - \log \mu)$, projection to $\mathsf{T}$ yields the stated expression, which coincides with $\nabla D_{\text{KL}}(\cdot \| \mu)$ up to a constant shift along $\mathbf{1}$ that vanishes. $\qquad\square$

[Thm. 2.7, constants] There exist $\delta > 0$ and $C > 0$ such that for $\beta \in [0, \delta]$, $\|(x_\beta^* - x_0^*,\, y_\beta^* - y_0^*)\| \leq C\beta$. One admissible constant is $C = \kappa M$ with $\kappa = \|J_0^{-1}\|$ and $M = \sup\{\|\partial_\beta g(x, y, \beta)\| : (x, y) \in \mathcal{N}, \beta \in [0, \delta]\}$ on a small neighborhood $\mathcal{N}$.

*Proof.* The implicit-function theorem gives $(x_\beta^*, y_\beta^*)$ continuously differentiable near 0 with derivative given by Lemma 5. Local Lipschitzness of $g$ in $(x, y, \beta)$ and boundedness of $J_0^{-1}$ imply $\|(x_\beta^*, y_\beta^*) - (x_0^*, y_0^*)\| \leq \int_0^\beta \|J_0^{-1}\|\, \|\partial_\beta g(x_s^*, y_s^*, s)\|\, ds \leq \kappa M\beta$. $\qquad\square$

**Bounding $\|\partial_\beta g\|$.** On $\mathsf{T}$, $\partial_\beta g(x, y, \beta) = (-(\log x - \log \mu), -(\log y - \log \mu))$. On a compact neighborhood with $x_i, y_j \geq \alpha > 0$, the map $p \mapsto \log p$ is $1/\alpha$–Lipschitz coordinatewise, hence $\|\partial_\beta g\| \leq c(\alpha, \mu)$.

**Norm equivalences used throughout.** On any compact subset of $\mathrm{relint}(\Delta_m)$, there exist constants $0 < \underline{\lambda} \leq \overline{\lambda} < \infty$ such that for all $v \in \mathsf{T}$,
$$\underline{\lambda} \|v\|_2^2 \ \leq \ v^\top \nabla^2 \psi(p)\, v \ \leq \ \overline{\lambda} \|v\|_2^2,$$
with $\nabla^2 \psi(p) = \mathrm{Diag}(1/p)$. This yields equivalence between Euclidean norms and the local information-geometric (entropic) norms and underlies the conversions between $\|u - u^*\|^2$ and $V$ in A.5.

**Summary.** A.1–A.6 establish: (i) the entropic Nash mirror ODE and interior invariance; (ii) local well-posedness; (iii) logit/quantal equilibria; (iv) a block Hamiltonian linearization with spectra $-\beta \pm i\sigma_k$; (v) Lyapunov decay and discrete linear rates with constants proportional to $\beta$; and (vi) $O(\beta)$ equilibrium sensitivity with explicit constants from $J_0^{-1}$ and Lipschitz moduli.

**Corollary A.3** (Parametric Taylor bound (specialization of Thm. 2.7))**.** *Let $\theta \mapsto (\pi_\theta, \pi'_\theta)$ be a $C^2$ local parametrization of policies near an interior unregularised NE $\theta_0^*$, and define $P_\theta := P(\pi_\theta \succ \pi'_\theta)$. Suppose the reduced Hessian $H_P(\theta_0^*) := \nabla_\theta^2 P_\theta\big|_{\theta_0^*}$ is positive definite on the tangent (with smallest eigenvalue $\lambda_{\min} > 0$). Then for sufficiently small $\beta$,*
$$\|\theta_\beta^* - \theta_0^*\| \ \leq \ \frac{\beta}{\lambda_{\min}} \big\|\nabla_\theta D_{\mathrm{KL}}(\pi_{\theta_0^*} \| \mu)\big\| \ + \ O(\beta^2),$$
*where $\theta_\beta^*$ is the $\beta$-regularised NE in parameter space.*

*Sketch.* Write the stationarity conditions for $P_\theta - \beta\, D_{\mathrm{KL}}(\pi_\theta \| \mu)$, subtract the unregularised condition at $\theta_0^*$, and first-order expand $\nabla_\theta P_\theta$ around $\theta_0^*$: $H_P(\theta_0^*)(\theta_\beta^* - \theta_0^*) = \beta\, \nabla_\theta D_{\mathrm{KL}}(\pi_{\theta_0^*} \| \mu) + O(\beta^2)$. Taking norms and using $\|H_P^{-1}\| = 1/\lambda_{\min}$ yields the claim. This is a parametric specialization of Theorem 2.7; see Appendix A.6 for the general IFT-based constants. $\qquad\square$

# B    PROOFS FOR SECTION 3

## B.1    PROOF OF PROPOSITION 3.1 (HESSIAN–$\beta$: INSTANTANEOUS DAMPING)

**Setup and notation.** Fix an iteration index $t$ and suppose the controller updates $\beta$ every $K \geq 1$ steps (Assumption C1). During the $t$-th primal step (or the $K$-step mini-epoch), $\beta$ is held constant at $\beta_t$; the dual/primal dynamics are those of the open-loop ODE with parameter $\beta_t$:
$$\dot{z} = G(z; \beta_t), \qquad z = (z_x, z_y), \quad x = \mathrm{softmax}(z_x), \ y = \mathrm{softmax}(z_y).$$
Let $J(z; \beta) := \nabla_z G(z; \beta)$ be the Jacobian of the dual field. At a neighborhood of the equilibrium $(x_{\beta_t}^*, y_{\beta_t}^*)$, Section 2 shows that
$$J(z^*; \beta_t) \ = \ \begin{bmatrix} -\beta_t I & H \\ -K & -\beta_t I \end{bmatrix}, \qquad \mathrm{spec}\big(J(z^*; \beta_t)\big) = \{-\beta_t \pm i\sigma_k\}_{k=1}^{m-1}, \qquad (8)$$
with $H = A J_y^*$, $K = A^\top J_x^*$, and $\{\sigma_k^2\}$ the nonzero eigenvalues of $HK$ (Theorem 2.4).

**Real parts (instantaneous damping).** Because $\beta$ is constant within the step, the linearization of the *closed-loop* system along the primal dynamics coincides with the *open-loop* linearization at $\beta_t$, hence equation 8 applies. Therefore, *instantaneously* the eigenvalues satisfy $\Re \lambda_t = -\beta_t$, proving the first claim.

**One-step lower bound on $\beta_{t+1}$.** By the controller update equation 5,
$$\beta_{t+1} \ = \ \mathrm{proj}_{[\beta_{\min}, \beta_{\max}]} \Big( (1 - \rho)\beta_t + \rho\, \zeta^*\, \hat{\sigma}_t \Big).$$
Assumption C2 gives $|\hat{\sigma}_t - \sigma_t| \leq \epsilon\, \sigma_t$ for a representative modal frequency $\sigma_t$. Hence $\hat{\sigma}_t \geq (1 - \epsilon)\sigma_t$ and
$$\beta_{t+1} \ \geq \ (1 - \rho)\beta_t + \rho\, \zeta^*(1 - \epsilon)\sigma_t$$
before projection; the projection can only increase this lower bound if it clips up to $\beta_{\min}$. This proves the inequality in Proposition 3.1.

**Uniform contraction rate.** If $\zeta^*(1 - \epsilon)\underline{\sigma} \geq \beta_{\min} > 0$ for a local lower bound $\underline{\sigma}$ of the representative modal frequency, then $\beta_{t+1} \geq \beta_{\min}$ holds inductively (assuming the projection box contains $\beta_{\min}$). Since $\Re\lambda_t = -\beta_t$, the local contraction rate of the linearised flow is uniformly bounded below by $\beta_{\min}$ at every step, completing the proof. $\square$

**Remark B.1** (Two time-scales and mid-epoch linearization). *Assumption C1 ensures $\beta$ varies on a slower time-scale than the primal dynamics. For $K > 1$, the* intra-epoch *linearization uses fixed $\beta_t$; for $K = 1$, small $\rho$ produces the same effect. Either way, the instantaneous real parts remain $-\beta_t$.*

B.2   Proof of Theorem 3.3 (Bias–$\beta$: budget tracking)

**Closed-loop map in** $\log\beta$. Let $u_t := \log\beta_t$. The controller equation 6 updates

$$u_{t+1} = u_t + \eta_\beta\,\delta_t, \qquad \delta_t = \text{clamp}\big(B(\pi_t) - B^*, -b, b\big). \tag{9}$$

Let $u^* = \log\beta^*$ with $B(\pi^*_{\beta^*}) = B^*$.

**Quasi-static approximation and linearization.** By two time-scales (C1) and local convergence (Theorem 2.6), the policy iterate tracks the regularised equilibrium: $\pi_t = \pi^*_{\beta_t} + e_t$ with a small tracking error $e_t$ that decays geometrically between controller updates. Write the budget map in the equilibrium proxy:

$$\bar{B}(u) := B(\pi^*_{e^u}), \qquad \bar{B}'(u^*) = s^* > 0 \quad \text{(Assumption C3)}.$$

Near $u^*$ and ignoring saturation ($|\delta_t| < b$), we have

$$\delta_t = \bar{B}(u_t) - B^* + \nu_t = s^*(u_t - u^*) + r_t + \nu_t,$$

where $r_t$ collects higher-order terms of the Taylor remainder (bounded by $c\,|u_t - u^*|^2$) and $\nu_t$ collects measurement/tracking noise from $e_t$ and mini-batching[1]. Substituting into equation 9 yields

$$u_{t+1} - u^* = (1 - \eta_\beta s^*)\,(u_t - u^*) + \eta_\beta(r_t + \nu_t), \tag{10}$$

until clamps activate. When clamps activate ($|\delta_t| = b$), the increment is deterministically bounded by $|\eta_\beta b|$ and we can treat saturation as an additional bounded disturbance.

**Mean stability and geometric rate.** Taking expectations and using $\mathbb{E}\,\nu_t = 0$ and $\mathbb{E}\,r_t = O(\mathbb{E}|u_t - u^*|^2)$ gives

$$\mathbb{E}[u_{t+1} - u^*] = (1 - \eta_\beta s^*)\,\mathbb{E}[u_t - u^*] + O\big(\mathbb{E}|u_t - u^*|^2\big).$$

For $0 < \eta_\beta s^* < 2$, the linear part has contraction factor $\phi := |1 - \eta_\beta s^*| < 1$. For small enough neighbourhoods (so that the quadratic term is dominated), this yields the claimed geometric rate $|\mathbb{E}[u_t - u^*]| \leq \phi^t |u_0 - u^*|$.

**Variance bound under bounded noise.** From equation 10, ignoring higher-order $r_t$ (or absorbing it into the disturbance as a bias term), we have an AR(1) recursion with bounded additive noise:

$$u_{t+1} - u^* = \phi\,(u_t - u^*) + \eta_\beta\xi_t, \qquad \phi = 1 - \eta_\beta s^*, \quad \xi_t := r_t + \nu_t,$$

where $|\xi_t| \leq b$ almost surely when clamps are active, and $\text{Var}(\xi_t) \leq \sigma_\xi^2 < \infty$ by C4 otherwise. Standard AR(1) variance calculus gives the stationary bound

$$\sup_t \text{Var}(u_t) \;\leq\; \frac{\eta_\beta^2\,\sigma_\xi^2}{1 - \phi^2} \;\leq\; \frac{\eta_\beta^2\,b^2}{1 - (1 - \eta_\beta s^*)^2},$$

where the second inequality uses the saturation envelope $|\xi_t| \leq b$ and thus $\sigma_\xi^2 \leq b^2$. This proves Theorem 3.3. $\square$

**Remark B.2** (On the sign and size of $s^*$). *Assumption C3 (strict increase in $u = \log\beta$) is natural when $B$ measures drift toward $\mu$ (e.g., length ratio, ECE), because Theorem 2.7 implies $\frac{d\pi^*_\beta}{d\beta}\big|_{\beta=0}$ points in the $\nabla D_{\mathrm{KL}}(\cdot\|\mu)$ direction; if $B$ correlates positively with this direction, then $s^* > 0$. In parametric models, this matches the first-order Taylor bound derived in the memo (Appendix A.6, Corollary A.3).*

---

[1]Under C4, $\nu_t$ is zero-mean with bounded variance; $e_t$ can be folded into $\nu_t$ due to geometric decay between controller steps.

### B.3 COMPOSITION: FEASIBILITY FIRST, DAMPING SECOND

We justify the priority composition

$$\beta_{t+1}^{\text{bias}} \text{ from equation 6}, \qquad \beta_{t+1}^{\text{hess}} \text{ from equation 5}, \qquad \beta_{t+1} = \max\{\beta_{t+1}^{\text{bias}}, \beta_{t+1}^{\text{hess}}\}.$$

**Proposition B.3** (Budget feasibility and damping lower bound). *Suppose C1–C4 hold. If the Bias–$\beta$ loop is mean-stable at $u^* = \log \beta^*$ (Theorem 3.3), then the composed update preserves budget feasibility in the limit: $B(\pi_{\beta_t}^*) \to B^*$ and $B(\pi_t) \le B^* + o(1)$ almost surely. Moreover, the realised damping obeys $\Re\lambda_t = -\beta_t \ge \min\{\beta_t^{bias}, \beta_t^{hess}\}$, and thus is at least the Hessian-target when the bias budget admits it (i.e., when $\beta_t^{hess} \ge \beta_t^{bias}$).*

*Proof.* By construction $\beta_{t+1} \ge \beta_{t+1}^{\text{bias}}$, so the composed loop cannot pick a $\beta$ smaller than the bias-feasible one. Under mean stability of Bias–$\beta$ (Theorem 3.3), $\beta_t^{\text{bias}} \to \beta^*$ and therefore $B(\pi_{\beta_t}^*) \to B^*$; two time-scales ensure $B(\pi_t) - B(\pi_{\beta_t}^*) \to 0$ in mean, yielding the stated feasibility. The damping statement follows from $\Re\lambda_t = -\beta_t$ (Proposition 3.1). $\square$

**Summary.** Hessian–$\beta$ sets the instantaneous damping via a spectral proxy, while Bias–$\beta$ enforces a budget in expectation; the max-composition attains the larger of the two $\beta$ values, ensuring *feasibility first, speed second*.

## C ADDITIONAL METHOD EXPLANATIONS

### C.1 DIAGNOSTICS, HYPERPARAMETERS, AND SAFEGUARDS

**Diagnostics.** (i) *$\beta$-sweeps:* plot $D_{\text{KL}}(\pi_\beta^* \| \pi_0^*)$, $\ell_2$, exploitability; (ii) *Spectral panels:* $\Re\lambda$ vs. $\beta$, $\Im\lambda$ (test Theorem 2.4); (iii) *Budget tracking:* $B(\pi_t)$ curves (overshoot and rate).

**Hyperparameters.** Hessian–$\beta$: $\zeta^* \in \{0.6, 0.8, 1.0\}$, $\rho \in \{0.2, 0.5\}$, update every $K \in \{1, 2\}$ steps. Bias–$\beta$: $\eta_\beta \in \{0.2, 0.4, 0.6\}$, $b \in \{0.25, 0.5\}$, $K \in \{1, 2, 4\}$. Bounds: choose $[\beta_{\min}, \beta_{\max}]$ from task ranges.

**Complexity.** Hessian–$\beta$: one JVP per controller step (two for centered differences); negligible vs. LLM forward. Bias–$\beta$: one scalar measurement on a small probe set. Memory overhead is negligible.

**Safeguards.** Two time-scales (small $\rho, \eta_\beta$ or larger $K$), projection $\beta \in [\beta_{\min}, \beta_{\max}]$ and clamping for anti-windup, EMA for noisy $\hat{\sigma}$, and periodic re-estimation of $\mu$ to avoid lock-in.

## D ADDITIONAL EXPERIMENTAL SETTINGS AND RESULTS

### D.1 EXPERIMENTS A: TOY GEOMETRY ON RPS

**Setup.** RPS payoff $A$ (antisymmetric), biased reference $\mu = (0.36, 0.32, 0.32)$. We implement Nash-MD/MP, APMD-like, MPO/GRPO-like, and the two controllers. We render vector fields (symmetric restriction $x = y$), simulate trajectories, sweep $\beta$, and compute Jacobian spectra numerically at fixed points.

**Hyperparameters.** Stepsizes $\eta \in [0.1, 0.2]$; grid step 0.08 for fields; 600 steps for trajectories; finite difference $h = 10^{-6}$ for spectra. All seeds and code are released (see Reproducibility).

**Findings.** (i) Vector fields and trajectories confirm *spiral steepening* with $\beta$; (ii) $D_{\text{KL}}(x_\beta^* \| x_0^*)$ increases with $\beta$ (bias amplification); (iii) real parts of spectra scale as $-\beta$, while imaginary parts remain roughly constant; (iv) Nash-MP and adaptive-$\beta$ controllers converge faster than fixed-$\beta$ MD while controlling bias.

**Quantitative comparison.** Table 2 summarises exploitability and budget-violation fractions (mean$\pm$CI over $n = 6$ random initializations).

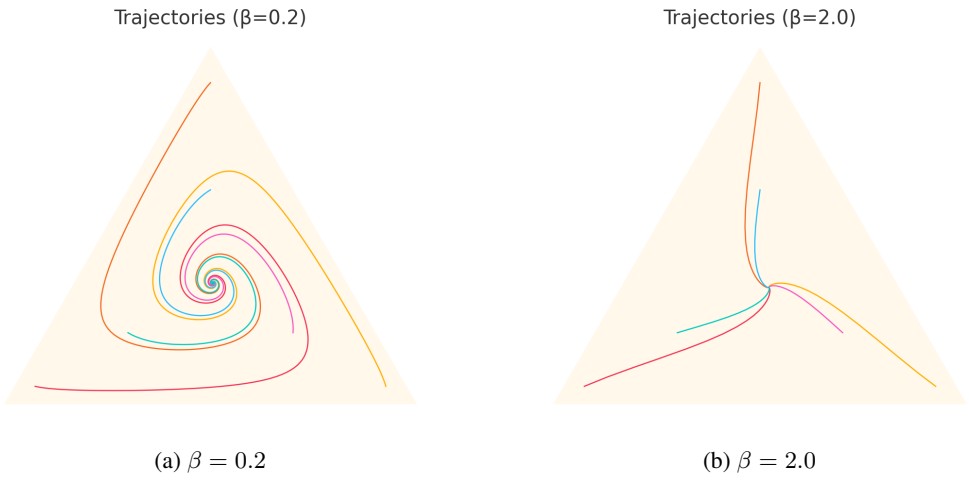

Mirror-descent vector field on $\Delta_2$ ($\beta$=0.0)   Mirror-descent vector field on $\Delta_2$ ($\beta$=1.0)

(a) Vector field ($\beta = 0$).  (b) Vector field ($\beta = 1$).

Figure 5: Simplex vector fields: larger $\beta$ yields steeper spirals (increased damping).

Trajectories ($\beta$=0.2)  Trajectories ($\beta$=2.0)

(a) $\beta = 0.2$  (b) $\beta = 2.0$

Figure 6: Trajectory bundles: faster radial decay for larger $\beta$.

### D.2 ABLATIONS AND DIAGNOSTICS

We ablate controller gains and $\beta$ values to stress the speed–bias trade-off.

**$\beta$-sweep.** $\beta \in \{0, 0.1, 0.3, 1.0, 3.0\}$. *Observed:* (i) bias to $\mu$ (KL, $\ell_2$) increases monotonically with $\beta$; (ii) exploitability decreases; (iii) spectral real parts follow $-\beta$ while imaginary parts are nearly constant. *Expected figures:* see Fig. 3 (right) for the toy setting; for LLMs we plan a small diagnostic panel mirroring these trends.

**Bias–$\beta$ gains.** $\eta_\beta \in \{0.2, 0.4, 0.6\}$, clamp $b \in \{0.25, 0.5\}$. *Observed:* larger $\eta_\beta$ improves budget tracking yet overshoots without clamps; clamps bound transients.

**Hessian–$\beta$ scaling.** Target ratios $\zeta^\star \in \{0.6, 0.8, 1.0\}$, relaxation $\rho \leq 0.5$. *Observed:* increasing $\zeta^\star$ accelerates convergence but may drift bias if $\hat{\sigma}$ is noisy; clamping $\hat{\sigma}$ and EMA smoothing mitigate this. *Expected figure:* step vs. exploitability for different $\zeta^\star$.

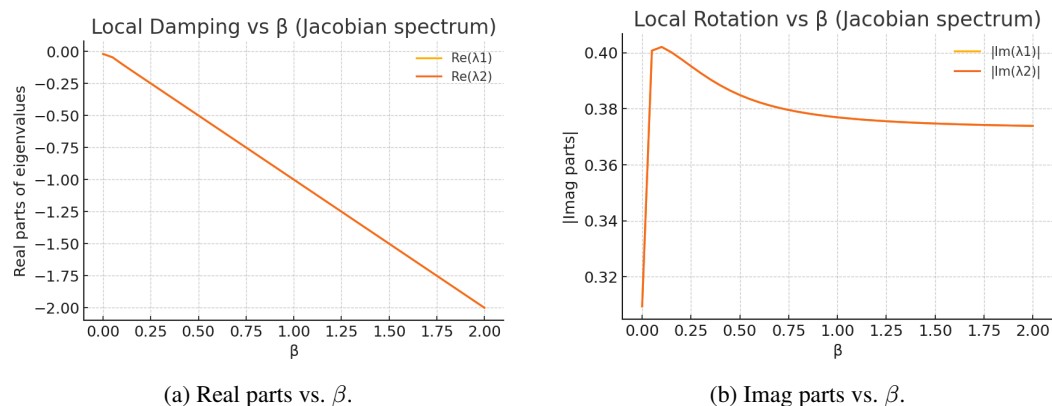

(a) Real parts vs. $\beta$.        (b) Imag parts vs. $\beta$.

Figure 7: Spectra: real parts track $-\beta$ (uniform damping); rotation stays near-constant.

Table 2: Toy RPS ($n=6$ inits). *Takeaway:* **Bias–$\beta$** satisfies the bias budget with low variance while keeping exploitability competitive; Adaptive-KL meets a KL target but violates the *operational* bias budget frequently.

| Method | final exploit (mean) | std | 95% CI | bias viol. frac (mean) | std | 95% CI |
|---|---|---|---|---|---|---|
| Nash–MD ($\beta=0.5$) | 0.0562 | 0.0000 | [0.0562, 0.0562] | 0.138 | 0.021 | [0.121, 0.155] |
| Nash–MP ($\beta=0.5$) | 0.0562 | 0.0000 | [0.0562, 0.0562] | 0.164 | 0.041 | [0.131, 0.197] |
| APMD-like ($\beta=0.5$, $\sigma=0.02$) | 0.0606 | 0.0000 | [0.0606, 0.0606] | 0.817 | 0.014 | [0.806, 0.829] |
| Adaptive–KL (KL$^*=0.03$) | 0.0398 | 0.0429 | [0.0054, 0.0742] | 0.814 | 0.059 | [0.767, 0.861] |
| **Bias–$\beta$** ($B^*=0.04$) | **0.0489** | 0.0005 | [0.0485, 0.0493] | **0.042** | 0.008 | [0.036, 0.048] |

**APMD noise.** $\sigma \in \{0, 0.01, 0.02\}$. *Observed:* noise breaks cycles but raises final exploitability at fixed targets. *Expected figure:* cycle magnitude vs. $\sigma$ and final exploitability vs. $\sigma$.

### D.3 REPRODUCIBILITY AND ARTIFACTS

We release anonymised code/notebooks/figures/eval scripts (https://anonymous.4open.science/r/geonash-263B/). We log harness versions and commit hashes, and fix decoding settings (temperature, top-p, max tokens). All hyperparameters, data splits, and seeds are documented. Toy/micro plots are reproducible on CPU/1×GPU in minutes; benchmark inference on 1×A100/H100 80GB completes within one day with SC(10) on GSM8K.

## E  THE USE OF LARGE LANGUAGE MODELS (LLMS)

The LLMs are used for assisting

- Manuscript structuring and editing
- Mathematical exposition (proof sketches and readability)
- LaTeX engineering

.

