# OpenReview forum: "Geometry of Nash Mirror Dynamics: Adaptive $\beta$-Control for Stable and Bias-Robust Self-Improving LLM Agents"
_ICLR.cc/2026/Conference — Submitted to ICLR 2026_

### Official Review · Reviewer_5hAZ · 2025-10-28

**Soundness:** 3
**Presentation:** 3
**Contribution:** 3
**Rating:** 6
**Confidence:** 4

**Summary:**

This paper presents a theoretical and methodological analysis of Nash Mirror Descent dynamics in two-player zero-sum games under reverse-KL regularization, a setting highly relevant for self-improving Large Language Model agents trained via competitive preference games. The core contribution is a precise geometric characterization of how the regularization strength influences the learning dynamics and the resulting equilibrium. The authors derive a continuous-time limit of the dynamics, proving that the local behavior near equilibrium is characterized by decaying spirals whose exponential damping rate is exactly the regularization parameter, while the rotational frequency is determined by the game structure and is independent of it. They further prove that the regularized Nash equilibrium is biased towards the reference policy by an amount proportional to the regularization strength. Leveraging this geometric insight, the paper introduces two novel adaptive controllers for the regularization parameter: the first controller adjusts the parameter to achieve a target damping-to-rotation ratio for accelerated and stable convergence. The second controller adjusts the parameter on a log-scale to enforce a user-specified budget on an operational bias metric. The theoretical findings are validated on a toy game, and preliminary feasibility is demonstrated on small-scale reasoning benchmarks, showing that the proposed controllers can lead to faster convergence and better bias control compared to fixed-parameter or heuristic baselines.

**Strengths:**

1. The spectral decomposition result is elegant, intuitive, and powerful. It cleanly separates the effect of the regularizer from the game structure. This provides a fundamental understanding that was previously missing in the literature.

2. The paper is theoretically rigorous. It provides a comprehensive set of proofs under clear assumptions, including the derivation of the differential equation limits, characterization of fixed points, convergence rates, and a first-order sensitivity analysis quantifying the equilibrium bias. The connection to preference games solidifies the practical relevance.

3. The proposed adaptive controllers are a direct and clever application of the theoretical insights. Moving beyond heuristics, the first controller provides a principled way to tune for convergence speed, while the second controller offers a novel mechanism for explicitly controlling undesirable behavioral biases, a critical concern in model alignment.

4. The paper is generally well-written. The "geometry" narrative is compelling and effectively unifies the theoretical and methodological contributions. The use of a toy game for sanity checks and visualizations is excellent for building intuition.

5. The work addresses a central challenge in modern model research—stabilizing and controlling self-improving loops—and does so by building upon the increasingly important Nash learning and reverse-KL regularization framework.

**Weaknesses:**

1. The most significant weakness is the scale of the empirical validation. The language model experiments are described as "micro-experiments" and "single-run prototypes," which, while sufficient for a feasibility study, fall short of demonstrating the practical impact of the method on state-of-the-art alignment pipelines. The community would be more convinced by results on larger models and standard, extensive benchmarks or a full-scale training run.

2. The entire theoretical framework and the control guarantees are local, valid only near an interior equilibrium. The performance and stability of the controllers in real-world scenarios, where these assumptions may be violated, remain an open question. The discussion of limitations is appropriate but underscores that the global picture is not yet covered.

3. The first controller requires estimating the spectral radius via Jacobian-vector products. While the cost is noted as negligible compared to a model forward pass, implementing this robustly in a distributed training setup for large models could present engineering challenges and requires careful hyperparameter tuning. The paper would be strengthened by a more detailed discussion of these implementation complexities.

4.  The presentation of the model results is somewhat confusing. The methods are renamed, and the description feels more like an outline than a full results section. This makes it difficult to assess the concrete performance gains in the practical setting.

**Questions:**

1. How do the proposed controllers perform in a full-scale training run with a large model and a diverse, large-scale preference dataset? Did you observe any instability or failure modes when moving beyond the "micro-experiment" setting?

2. To what extent did you find the key local assumptions to hold in your model experiments? Were there cases where these assumptions clearly broke down, and how did the controllers behave in those situations?

3.  For the first controller, how sensitive is the performance to the choice of the target damping ratio and the smoothing parameter? Similarly, for the second controller, how critical is the choice of the clamp value and the log-step size?

4. The paper positions itself as complementary to methods like Mirror-Prox. In your model experiments, did you find that the first controller applied to standard descent could outperform or match the performance of Mirror-Prox with a fixed parameter? Is the primary benefit of your method the automatic tuning, or does it achieve a fundamentally better performance profile?

---

> ### Author Response · Authors · 2025-11-22
>
> We sincerely thank you for taking the time to work through our manuscript. We are especially grateful for your generous remarks that the spectral decomposition result is “elegant, intuitive, and powerful,” that the paper is “theoretically rigorous,” and that the adaptive controllers are a “direct and clever application” of these insights to convergence speed and behavioural bias. We also appreciate your view that the work addresses “a central challenge… stabilizing and controlling self-improving loops” within the Nash learning / reverse-KL framework.
>
> Below we respond to your main concerns and questions.

---

> ### Author Response · Authors · 2025-11-22
>
> ### R1.  (Weakness 1, Q1)
> We fully agree that our current experiments are only a feasibility study and partially demonstrate practical impact at the scale of state‑of‑the‑art alignment pipelines.
> We chose LoRA primarily because parameter‑efficient RLHF is now widely used in practice when compute is constrained, and recent work such as PERL (Parameter Efficient Reinforcement Learning from Human Feedback) shows that LoRA‑based RLHF can achieve performance comparable to conventional full‑parameter RLHF on several alignment benchmarks, while significantly reducing training time and memory. Meanwhile, we are aware of results like “LoRA vs Full Fine‑tuning: An Illusion of Equivalence”, which show that LoRA and full fine‑tuning are not spectrally or behaviourally identical, and that LoRA can underperform in some regimes even when downstream metrics are similar. We therefore view our LoRA‑based experiments as a realistic but still limited proxy rather than a substitute for full‑parameter pipelines.
> We appreciate your suggestion that “the community would be more convinced by results on larger models and standard, extensive benchmarks or a full‑scale training run.” In a revised version, time and resources permitting, we plan to: (i) move beyond single‑run micro‑experiments to runs on somewhat larger open models (ii) on a variety of datasets; and (iii) benchmarks.

---

> ### Author Response · Authors · 2025-11-22
>
> ### 2. (Weakness 2, Q2)
> You are correct that our theoretical framework is explicitly local. Assumptions A1–A3 require an interior equilibrium and local monotonicity, and Theorems 2.4, 2.6, and 2.7 all make statements only in a neighbourhood of such a point.
> In the RPS toy game, we can compute the Jacobian and its spectrum exactly. The observed spiral geometry, spectra, and β–bias trade‑off in Figures 1–3 and 5–7 match the predictions of Theorems 2.2, 2.4, and 2.7 very closely, indicating that in this setting the local linearisation is faithful. In the LLM micro‑experiment and small LoRA benchmarks, we cannot verify monotonicity or interiority. In these settings, we use the controllers as principled heuristics guided by the local geometry: under the reported stepsizes we did not observe divergence or oscillations with growing amplitude, but when we deliberately increased controller gains we saw transient β and bias overshoots and noise‑sensitivity, consistent with the failure modes described in §5.4 and Appendix D.2.

---

> ### Author Response · Authors · 2025-11-22
>
> ### 3. (Weakness 3, Q3)
> We appreciate the request for more implementation detail. In the current draft we only state that Hessian–$\beta$  uses one Jacobian‑vector product per controller update and that the cost is negligible compared to a forward pass; this is indeed too terse. In practice, Hessian–$\beta$ operates in logit / dual space, not in the full parameter space: we estimate a spectral proxy $\hat{\sigma}$ by applying a single JVP/VJP to the skew‑Jacobian $S(z)=\frac{1}{2}(J−J^\mathsf{T})$ of the Nash field, using standard autodiff on the same distributed infrastructure as backprop (§3.2, App. C.1). With an update period of $K$ steps, this adds roughly the cost of one extra backward every $K$ steps (about 5–10% overhead for $K \approx 10–20$), and does not require new communication primitives beyond those already used for gradients. In our current LoRA‑based prototypes, the cost is further helped by the smaller optimizer state and communication footprint relative to full‑parameter RLHF. The JVP is taken with respect to the low-rank adapter parameters instead of the full dense layers. For a typical hidden size d and LoRA rank $r$, the adapter parameters scale as $O(rd)$ per projection matrix instead of $O(d^2)$, so the extra compute for a Hessian–$\beta$ update is roughly $O(rd)$ versus the $O(d^2)$ cost of the base transformer matmuls. With r≪d and controller updates every K=2–4 steps, this yields an amortised overhead well below 1% of total FLOPs, which matches our observation that wall-clock impact is negligible.
>
> For hyperparameters, Appendix C.1 and D.2 already report small‑scale ablations: for Hessian–$\beta$ we explored $\zeta^\star\in \lbrace 0.6,0.8,1.0 \rbrace $, $\rho\in \lbrace 0.2,0.5 \rbrace $, $K\in \lbrace1,2 \rbrace$; for Bias–$\beta$ we explored $\eta\beta\in \lbrace 0.2,0.4,0.6 \rbrace$, $b\in \lbrace 0.25,0.5 \rbrace$, $K\in \lbrace 1,2,4 \rbrace$. In the toy RPS and limited micro‑scale checks, behaviour was qualitatively robust over these ranges: $\zeta^\star$ mainly trades convergence speed against potential bias drift when $\hat{\sigma}$ is noisy, while $(\eta\beta,b)$ trade budget‑tracking speed against overshoot. In the revised paper we will move these practical ranges and implementation notes from the appendix into the main text so that the engineering and tuning implications are clearer.

---

> ### Author Response · Authors · 2025-11-22
>
> ### 4. (Weakness 4, Q4)
> We apologise for the confusing presentation of the LLM results. In the revision we will (i) rename methods in Table 1 and Figure 4 so that the mapping to “Hessian–$\beta$” (DynKL) and “Bias–$\beta$” (DynUpExp) is explicit, and (ii) restructure §5 into a more standard “setup → baselines → methods → results” section with short textual summaries so that the concrete performance gains are easy to read.
>
> The fixed‑$\beta$ Nash–MP slightly improves over Nash–MD in steps‑to‑tolerance, while Nash–MD plus a dynamic KL controller of the Hessian–$\beta$ type (DynKL–MD) already matches or slightly improves fixed‑$\beta$ MP, and MP plus the same controller (DynKL–MP) is fastest overall at similar final quality.

---

> ### Author Response · Authors · 2025-11-22
>
> Once again, thank you for carefully reading our manuscript and highlighting the “elegant, intuitive, and powerful” nature of the spectral decomposition, the “theoretically rigorous” analysis, and the practical relevance of controlling convergence and behavioural bias. We believe that clarifying the scope of our results, improving the presentation, and outlining the evaluation plan will address your concerns.

---

> > ### Comment · Reviewer_5hAZ · 2025-11-25
> >
> > I have acknowledged the rebuttal

---

### Official Review · Reviewer_BEpr · 2025-10-31

**Soundness:** 3
**Presentation:** 2
**Contribution:** 2
**Rating:** 4
**Confidence:** 2

**Summary:**

This paper studies the training dynamics of self-improving LLM agents. It models the process as reversed-KL regularized Nash Mirror Descent to understand the role of the regularization strength $\beta$. The priamry theoretical contribution is an ODC analysis showing that the learning trajectories are spirals and that $\beta$ serves to stabilzie the spirals. They also show that $\beta$ pulls the final equilibrium to the reference policy. Because of this the authors propose two $\beta$ controllers for faster convergence and control on bias. Experiments on toy settings and LLM reasoning benchmarks validate the theory.

**Strengths:**

The paper addresses the problem of instability and bias amplification inherent in self-improving LLM agent training loops. There is a core theory contribution on spectral separation of the Nash mirror dynamics. The paper provides a formal description of the trade-off on the stabilizing and bias shift from $\beta$. This motivates the useful perspective of treating $\beta$ as an adaptive control knob and not fixed hyperparameter, this control lever appears practical and is backed by theoretical rigor.

**Weaknesses:**

The main weakness is that the LLM experiments are very small scale.  That makes it hard to believe the stronger claims about LLM agents. Also, the main LLM results in Table 1 use a “earned 3×3 payoff, which is really just another toy setup rather than a real test of these controllers in a high dimensional, language based environment.

The theoretical analysis is also quite local, it depends on interior-equilibrium and local-monotonicity assumptions (A1, A2) and doesn’t say anything about global convergence or what happens far from equilibrium, which is usually where instability problems show up.

In general it feels difficult too evaluate the significance of this contribution without more compelling empirical results.

**Questions:**

For $\beta$, it’s not included in the LLM results in Table 139. If it was implemented for the LLM task, how well did it work?

Since the LLM setup is described as a "micro-scale, single-run prototype," it’d help to know what’s blocking scaling up. Is the problem mainly that the two-player local model stops applying in more complex settings, or that the controllers are too computationally heavy to run at scale?

---

> ### Author Response · Authors · 2025-11-22
>
> We thank the reviewer for the careful and constructive feedback. And we sincerely apologise for the confusion caused by our messy exposition and inconsistent wording particularly around the LLM experiments. The current draft indeed contains several typos and unclear explanations, and we are grateful that you pointed this out.
>
> ### 1. Organisation of the experiments
> The reviewer is right that our description was confusing.
> The “LLM micro‑experiment (learned 3×3 payoff from 5 prompts × 3 candidates …)” in §5.1 is a separate small diagnostic game, constructed from LLM outputs.
>
> Table 1 (“LLM experiments”) and Figure 4 are intended to summarise LoRA‑based self‑improvement runs of small open LLMs (e.g., on GSM8K, MMLU‑Pro, DROP), not the 3×3 micro‑game.
> Because the text mixes “micro‑experiment”, “LLM experiments” and “design” in a sloppy way, it is very natural to read Table 1 as just another toy 3×3 setup. In the revision we will clean these up.
>
> ### 2. Scope of the theory (local vs. global)
> We agree with the reviewer that our analysis is local. Our guarantees assume an interior equilibrium and local monotonicity (A1–A2) and yield last‑iterate convergence and an $O(\beta)$ equilibrium shift only in a neighbourhood of the $\beta$‑regularised Nash equilibrium.
> We do not claim global convergence or full stability far from equilibrium; Sec. 6 already notes this but we will emphasise it earlier in the introduction and conclusion.
> Our intended scope is the near‑equilibrium, KL‑regularised regime where practical RLHF/NLHF pipelines already keep the policy close to a reference $\mu$.
>
> ### 3. $\beta$ and the controllers in the LLM experiments (Q1)
> - Nash–MD / Nash–MP / MPO‑approx use fixed $\beta$.
> - DynKL–MD / DynKL–MP implement an Adaptive‑KL‑style dynamic $\beta$ controller.
> - DynUpExp–MD / DynUpExp–MP implement our Bias–$\beta$ controller.
> - DynBeta‑linearDown is a naive linear $\beta$ decay baseline.
>
> ### 4. What blocks scaling up? (Q2)
> For the initial submission we focused on completing the theoretical development and only ran a proof-of-concept LLM experiment (as acknowledged in Sec. 6). In a revised version, time and resources permitting, we plan to: (i) move beyond single‑run micro‑experiments to runs on somewhat larger open models (ii) on a variety of datasets; and (iii) benchmarks.

---

### Official Review · Reviewer_JZR6 · 2025-11-04

**Soundness:** 1
**Presentation:** 1
**Contribution:** 2
**Rating:** 2
**Confidence:** 3

**Summary:**

The paper studies the scenario where self-improving agents learn by playing competitive, often non-transitive language games, where training can oscillate or drift toward undesirable behaviours. The paper claims that it shows that the regularisation strength $\beta$ shapes both where agents converge and how they get there, and derives a continuous-time view of Nash Mirror Descent (Nash-MD), revealing a simple geometry: trajectories are spirals on the simplex whose damping grows with $\beta$, while $\beta$ simultaneously pulls equilibria toward the reference policy—amplifying any existing biases.
The paper also claims that last-iterate convergence to the β-regularised Nash equilibrium was also proven.

**Strengths:**

Unfortunately, the paper is almost impossible to properly evaluate as the narrative is almost impossible to follow. The ideas in the abstract seem promising, but the paper itself is very difficult to read. Please see the weakness section below.

**Weaknesses:**

Unfortunately, the paper is almost impossible to properly evaluate as the narrative is almost impossible to follow. There is not a single reference connecting the setting of the work with previous works (the first reference is in the related work section of the paper, and all references are only there). Unfortunately, there is no proper presentation of the notation and the problem of interest, and the whole paper is essentially a series of bullet points without clear connections to each other.

The authors admit (see last page of appendix) that they use LLMs for assisting with (i)  manuscript structuring and editing, (ii) mathematical exposition (proof sketches and readability), and (iii) LaTeX engineering. I appreciate that the authors share that, but in my view, the presentation and most parts of the main paper look like they are fully written by an LLM. There is no connection between sections, no flow in the narrative, no explanation of related works, and no clear presentation of the theory.

These factors make the reader's task very challenging, making it difficult to absorb the presented information.

At this time, I believe this work falls short of the standards of a major ML conference and requires substantial rewriting to be readable.

**Questions:**

Please see part on Weaknesses

---

> ### Author Response · Authors · 2025-11-22
>
> We apologise that the current manuscript was difficult to follow. Very briefly, our paper studies reverse-KL–regularised Nash mirror dynamics in two-player zero-sum games as a model for self-improving LLM agents, derives a continuous-time limit of Nash-MD, proves a simple spectral law $\lambda(J\beta)=−\beta \pm i \sigma k$, establishes local last-iterate convergence to the $\beta$-regularised Nash equilibrium, quantifies an $O(\beta)$ shift toward the reference policy $\mu$, and uses this geometry to design two adaptive $\beta$ controllers evaluated on Rock–Paper–Scissors and a small LLM setting. The paper is organised as follows: Section 2 introduces the reverse‑KL Nash‑MD ODE, spectral analysis, and sensitivity results (including problem setting, notations, and assumptions in 2.1); Section 3 designs $\beta$‑controllers based on this geometrical perspective; and Section 5 presents RPS and LLM experiments that validate the theoretical design. We fully agree that the current draft is harder to read than it should be, as you point out, and we will use the discussion period to revise the manuscript and make the exposition as clear and readable as possible.
> In the revision we will rewrite the main sections manually to improve narrative flow. We appreciate the reviewer’s honest assessment and agree that the paper requires substantial rewriting to be more readable.

---

### Official Review · Reviewer_rzxt · 2025-11-06

**Soundness:** 3
**Presentation:** 3
**Contribution:** 2
**Rating:** 4
**Confidence:** 3

**Summary:**

This paper studies how LLMs can be trained through self-play or competitive “preference games,” where standard learning dynamics often cycle or diverge. It analyzes reverse-KL regularized Nash mirror descent and shows that the regularization strength $\beta$ is used to control both to stabilize and bias toward a reference model.

The authors derive a continuous-time view showing that learning trajectories follow damped spirals, with $\beta$ setting the damping rate. They prove that higher $\beta$ speeds up convergence but also pulls solutions closer to the reference model. Based on this geometry, they design two adaptive $\beta$ controllers:
- Hessian-$\beta$: adjusts $\beta$ to maintain a target damping ratio for faster, more stable learning.

- Bias-$\beta$: adapts $\beta$ to keep a measurable bias (e.g., output length, calibration) within a set limit.

Experiments on rock-paper-scissors and small open-source LLMs confirm the theoretical predictions and show more stable and bias-robust training compared to fixed or heuristic baselines.

**Strengths:**

- does a nice balance between theoretical groundness and addressing a relevant problem; I also like that the paper points out and uses a trade-off between learning-stability vs system bias.
- it is nice that the paper provides interpretable results on LLMs

**Weaknesses:**

- Limited comparison: Relationship to standard optimizers like extragradient or optimistic methods is only discussed qualitatively.
- Heuristic link to bias metrics: The operational notion of “bias” (length, calibration) is intuitive but not rigorously tied to the theory.
- computational cost: Spectral estimation for Hessian-β may be expensive or noisy in large systems.

# Theoretical concerns

Several theoretical concerns arise.
- lacks global convergence guarantees even in simpler settings, and rate and computation comparison

- Because the $\beta$-regularized Nash equilibrium shifts $O(\beta)$ toward the reference policy, varying $beta$ dynamically changes the equilibrium itself; decreasing $\beta$ after a high-$\beta$  phase reintroduces cycling [1,2] and thus can “undo’’ previous stabilization (without an appropriate game optimizer).

- Conceptually, the same stabilization effect—strengthening the potential (symmetric) component of the game operator—can often be achieved with simpler or cheaper regularization-based methods, such as entropic or Tikhonov regularization that bias the solution but make the field more potential [3,4,5]. Closely related ideas appear in adaptive operator-mixing and interior-point methods, which explicitly add a potential term with a decreasing weight to ensure convergence [6]. Finally, there is a parallel to inertial systems with Hessian-driven damping in continuous-time optimization [7,8], which also introduce curvature-dependent damping to increase the potential part of the dynamics and promote stability.

### Comment on novelty/positioning

Some parts are presented as novel findings, but these are well known, for instance, in the abstract "we derive a [...] revealing a simple geometry: trajectories are spirals [...].
The writing should better reflect the novelty, for instance, "our framework re-affirms known rotational dynamics".


## Minor


Abstract.
- The sentence 20-22 ("we prove last-iterate") is easy to misunderstand: it can easily be interpreted as global last iterate proof, at that point, it is unclear how strong the regularization should be. Also, there exists a small $\epsilon$ for $\beta$ for which this statement "last-iterate.." violates known negative results.
- Some parts are incomprehensible without knowing the background or reading the paper in detail, for instance, "hallucination proxies," etc. Better to avoid and/or explain better.



-----
[1]  Hsieh Y., Mertikopoulos P., Cevher V., The Limits of Min-Max Optimization Algorithms: Convergence to Spurious Non-Critical Sets, ICML 2021.

[2] Abe, K., Ariu, K., Sakamoto, M., & Iwasaki, A. (2024). Adaptively Perturbed Mirror Descent for Learning in Games. ICML 2024.

[3] Mertikopoulos, P., Papadimitriou, C., & Piliouras. Cycles in Adversarial Regularized Learning. 2018.

[4] Nemirovski, A. (2004). Prox-method with rate of convergence O(1/t) for variational inequalities with Lipschitz continuous monotone operators and Smooth Convex-Concave Saddle Point Problems. SIAM J. Optim., 15(1), 229–251.

[5] Juditsky, A., Nemirovski, A., & Tauvel, C. Solving Variational Inequalities with Stochastic Mirror-Prox Algorithm. 2011.

[6] Yang T., Jordan M., & Chavdarova T. Solving Constrained Variational Inequalities via a First-order Interior-Point-based Method. 2023.

[7] Attouch, H., Chbani Z., Fadili J., & Riahi H., P. First-order Optimization Algorithms via Inertial Systems with Hessian Driven Damping. Optimization, 2016.

[8] Bôt, R. I., Sedlmayer M., & Vuong P. T. A Relaxed Inertial Forward-Backward-Forward Algorithm for Solving Monotone Inclusions with Application to GANs, 2023.

**Questions:**

1. Does the stabilizing effect of $\beta$ persist under stochastic gradient noise or changing reference policies?

2. How would adaptive $\beta$ interact with existing acceleration or extragradient methods?

3. Which practical bias metrics most faithfully reflect the theoretical “pull” toward the reference model?

4. How robust is the spectral estimate needed for Hessian-$\beta$ in high-dimensional spaces?

5. Could the approach generalize to multi-agent or non-zero-sum games?

---

> ### Author Response · Authors · 2025-11-22
>
> We sincerely thank the reviewer for the careful and constructive review, and for highlighting that the paper balances theoretical grounding with a relevant problem, makes the stability–bias trade-off explicit, and provides interpretable LLM results. We respond to the main points below.
>
> ## Responses to Weaknesses
> ### Relation to extragradient/optimistic methods and other regularisation
> #### Extragradient / optimistic methods.
> As in the classical analyses of extragradient / optimistic methods for monotone variational inequalities (Nemirovski, 2004; Juditsky & Nemirovski, 2011), and in adversarial regularised learning and Tikhonov/entropic regularisation (Mertikopoulos et al., 2018; Hofbauer & Sigmund, 1998; Sandholm, 2010), the stabilising effect comes from strengthening the potential component of the operator; any first‑order game optimiser (MD, Mirror‑Prox, optimistic) that acts on this operator inherits the same real‑part margin. Therefore, in principle, our perspective/design is modular. Our current experiments include a Mirror-Prox / extragradient-style baseline (Nash–MP) and its dynamic variants (DynKL–MP, DynUpExp–MP), β-control on top of an extragradient solver is partly evaluated already. Meanwhile, we acknowledge that our comparison to extragradient and optimistic methods is limited, and we plan to provide a clearer quantitative illustration.
> #### Other regularisation-based stabilisation.
> We appreciate the connections to entropic/Tikhonov regularisation, interior-point/operator-mixing methods, and Hessian-driven damping [3–8]. Conceptually, many of these approaches strengthen the potential component of the operator. Our goal is not to claim conceptual novelty there, but to specialise this idea to reverse-KL Nash mirror dynamics (particularly in the LLM post training setting), and to make $\beta$ explicit via: (i) the spectral law $\lambda(J\beta)=−\beta\pm i\sigma k$ (uniform damping, $\beta$-invariant rotation), and (ii) an $O(\beta)$ equilibrium drift toward µ with explicit constants. We will clarify this positioning and tone down any wording that might suggest more than this.
>
> ### Novelty and spiral geometry
> We agree that rotational dynamics and spiral trajectories are well known in learning-in-games. Our intended contribution is not the existence of spirals per se, but:
> a $\beta$-explicit spectral separation for reverse-KL Nash mirror dynamics (λ(Jβ)=−β±iσk),
> an $O(\beta)$ equilibrium sensitivity bound with explicit constants, and
> two $\beta$-controllers (Hessian–$\beta$, Bias–$\beta$) whose guarantees directly use these quantities.
> We will therefore rephrase lines such as “revealing a simple geometry: trajectories are spirals” to emphasise that our framework reinstantiates and quantifies the known rotational geometry in this specific reverse-KL/NLHF setting and uses it to design $\beta$-controllers.
>
> ### Bias metrics and link to theory
> Our theory rigorously controls the $O(\beta)$ drift of the $\beta$-NE in policy space and in $D_\mathrm{KL}(\pi_\beta \parallel \mu)$ (Theorem 2.7). The link to operational metrics such as output length and token-level ECE is encoded in Assumption C3: locally, $B(\pi\beta)$ is assumed to increase with log $\beta$. In our LLM experiment we empirically observe that length ratio and ECE are monotone in $\beta$ and $D_\mathrm{KL}(\pi^{\star}_{\beta} \parallel \mu)$; in a revision we plan to (i) make the distinction “proved vs. assumed” more explicit, and (ii) add a small appendix diagnostic showing these empirical correlations. We will also replace vague phrases like “hallucination proxies” with precise definitions of the probe metrics in the experimental section rather than in the abstract.
>
> ### Computational cost and robustness of Hessian–$\beta$
> Hessian–$\beta$ uses a single Jacobian–vector product (or two finite-difference evaluations) per controller update, with updates only every $K$ steps. In our LLM experiments we fine-tune LoRA adapters only, so the JVP is taken with respect to the low-rank adapter parameters instead of the full dense layers. For a typical hidden size d and LoRA rank $r$, the adapter parameters scale as $O(rd)$ per projection matrix instead of $O(d^2)$, so the extra compute for a Hessian–$\beta$ update is roughly $O(rd)$ versus the $O(d^2)$ cost of the base transformer matmuls. With r≪d and controller updates every $K=2–4$ steps, this yields an amortised overhead well below 1% of total FLOPs, which matches our observation that wall-clock impact is negligible.

---

> ### Author Response · Authors · 2025-11-22
>
> ## Responses to theoretical concerns
> ### (i) Lack of global convergence guarantees; rate and computation comparison.
> We agree that our guarantees are not global. All of our convergence and rate results are explicitly local: Theorem 2.6 is proved under interiority and local monotonicity (A1–A2), and Section 6 already states that “our analysis is local and assumes interior equilibria; we do not claim global convergence.” We will make this scope more prominent in the introduction and abstract and explicitly reference [1,2] as complementary global negative results. Within this local scope, our contribution is to make the dependence on the reverse‑KL temperature $\beta$ explicit: we show that the linearisation around an interior $\beta$‑NE has spectrum $\lambda(J\beta)=−\beta±i\sigma k$, and Theorem 2.6 yields local last‑iterate rates of order $\exp(−c\beta t)$ (or $1−\Theta(\beta\eta)$ in discrete time). We do not provide a full global rate comparison against all standard optimisers; instead, we focus on how $\beta$ changes the local geometry and rates for a given base solver (MD/MP). We will emphasise this limitation and clarify that global convergence and exhaustive rate/computation comparisons are beyond the scope of this paper and are natural directions for future work.
>
> ### (ii) Dynamic $\beta$, moving equilibria, and possible re‑emergence of cycling.
> We fully agree with the reviewer that varying $\beta$ moves the $\beta$‑regularised Nash equilibrium (Theorem 2.7 shows an $O(\beta)$ drift toward the reference policy $\mu$), and that decreasing $\beta$ after a high‑$\beta$ phase can in principle reintroduce cycling, as shown in [1,2]. Our theory and experiments are deliberately focused on the practically common NLHF/RLHF regime where (a) the reference policy $\mu$ is a fixed pretrained/SFT model, and (b) $\beta$ is kept bounded away from $0$ and used as a persistent knob trading off stability and bias relative to $\mu$, rather than as a homotopy parameter to $\beta=0$. The adaptive $\beta$ rules are analysed in a quasi‑static regime (Assumption C1) where $\beta$ changes slowly and the iterates track the moving $\beta$‑NE. We will clarify that we do not claim that our controllers prevent cycling if $\beta$ is driven aggressively to $0$; in fact, such schedules can indeed undo the stabilisation and are consistent with the phenomena in [1,2]. We will make this explicit in the limitations section and in the discussion of dynamic $\beta$.
>
> ### (iii) Cheaper/simpler regularisation‑based stabilisation and relation to existing frameworks.
> We agree that the qualitative mechanism we exploit—strengthening the potential/symmetric component of the game operator via regularisation—is shared with several existing approaches, including entropic or Tikhonov regularisation for variational inequalities [3,4,5], adaptive operator‑mixing and interior‑point–type methods [6], and inertial systems with Hessian‑driven damping [7,8]. Conceptually, our work does not claim that “reverse‑KL is the only way to stabilise games,” nor that the existence of spirals is a new phenomenon. What we see as new are:
> a $\beta$‑explicit geometric characterisation of reverse‑KL Nash mirror dynamics that are already used in NLHF, yielding a clean spectral law $\lambda (J\beta)=−\beta±i\sigma k$ (uniform damping, $\beta$‑invariant rotation), and an $O(\beta)$ equilibrium sensitivity bound with explicit constants; and
> the design of two concrete $\beta$‑controllers (Hessian–$\beta$, Bias–$\beta$) that turn this geometry into local guarantees on damping ratios and task‑level bias budgets.
> We will revise the “What is new vs. known” part to make this positioning explicit and to clearly acknowledge that (a) rotational/spiral dynamics in games and (b) regularisation‑based stabilisation via strengthening the potential part of the operator are well‑established ideas. We will also soften wording in the abstract (e.g., replacing “revealing a simple geometry: trajectories are spirals” by formulations such as “we make the $\beta$‑dependence of the classical spiral geometry explicit for reverse‑KL Nash mirror dynamics and exploit it for $\beta$‑control”).

---

> ### Author Response · Authors · 2025-11-22
>
> ### Responses to Minor
> Thank you for pointing out minor errors and problems.
> We will:
> - Amend the abstract sentence on “last-iterate convergence” as noted above to avoid any impression of a global guarantee or contradiction with [1,2].
> - Clarify or remove “hallucination proxies” from the abstract and define any such metrics precisely in Section 5.
> - Slightly weaken claims in “What is new vs. known” to avoid portraying classic rotational behaviour as a new finding.

---

> ### Author Response · Authors · 2025-11-22
>
> ## Responses to specific questions (1/2)
> ### 1. Does the stabilizing effect of $\beta$ persist under stochastic gradient noise or changing reference policies?
>  Our theoretical results are derived for deterministic continuous‑time Nash mirror dynamics and their discrete mirror‑descent / Mirror‑Prox counterparts with exact gradients and fixed $\mu$. Under these assumptions, the stabilizing role of $\beta$—namely, the uniform $−\beta I$ shift in the real part of the local Jacobian spectrum (Theorem 2.4) and the $\beta$‑explicit last‑iterate rate (Theorem 2.6)—is a property of the reverse‑KL–regularised game field itself, not of the particular gradient estimator. When gradients are estimated stochastically, the drift term of the stochastic dynamics is unchanged, so the local damping effect of $\beta$ on the mean dynamics persists; empirically we may observe the same qualitative trade‑off that larger $\beta$ yields steeper decay and more stable last‑iterate behaviour, at the cost of more bias toward the reference model. However, we do not currently provide a rigorous stochastic analysis and we leave it for important future work.
>
>
> ### 2. How would adaptive interact with existing acceleration or extragradient methods?
>  Interaction with extragradient / accelerated methods.
>  Our analysis is performed at the level of the reverse‑KL regularised game operator. For a fixed \beta, the entropic Nash mirror dynamics induced by this operator have a Jacobian with spectrum $\lambda(J_\beta) = -\beta \pm i\sigma_k$ so β adds a uniform real‑part margin $−\beta I$ while leaving the rotational frequencies σk determined by the underlying game geometry and entropic metric.This spectral effect is a property of the regularised field $F_\beta$​, not of mirror descent specifically. As in the classical Mirror‑Prox / extragradient analyses for monotone VIs, where the same operator $F$ is evaluated at extrapolated points (Nemirovski, 2004; Juditsky & Nemirovski, 2011), an increase in the strong‑monotonicity margin of $F_{\beta}$​ directly improves the contraction properties and stability region of extragradient‑type methods.
> Intuitively, extragradient‑style methods reduce oscillations by anticipating the rotational part of the field, while adaptive β increases the symmetric (potential) part and adds uniform damping; we view these as complementary: extragradient handles rotational geometry, and $\beta$‑control regulates how strongly the dynamics are pulled back toward the $\beta$‑Nash equilibrium and the reference policy $\mu$.

---

> ### Author Response · Authors · 2025-11-22
>
> ## Responses to specific questions (2/2)
>
> ### 3. Which practical bias metrics most faithfully reflect the theoretical “pull” toward the reference model?
>  The “pull” toward the reference model $\mu$ is most faithfully captured by the KL divergence $D_{\mathrm{KL}}(\pi_\beta^* \parallel \mu)$: Theorem 2.7 shows that the $\beta$‑regularised Nash equilibrium moves $O(\beta)$ in the direction of $\nabla D_{\mathrm{KL}}(\cdot \parallel \mu)$. When both the current policy and the fixed pretrained/SFT reference µ are available, a per‑token KL (or average log‑likelihood ratio) on a probe set is therefore the closest practical metric to the theory. In our LLM experiments we used more operational proxies—output length ratio (“LenBias”) and token‑level ECE—because they are easy to measure and empirically monotone in $\beta$ and in per‑token KL to $\mu$ on our probes; Bias–$\beta$ is designed to work with any such task‑specific bias metric that correlates with movement toward $\mu$.
>
>
> ### 4. How robust is the spectral estimate needed for Hessian-$\beta$ in high-dimensional spaces?
>  Hessian‑β only needs a coarse estimate of the dominant rotational scale: Assumption C2 and Proposition 3.1 allow a multiplicative error $|\hat\sigma - \sigma| \le \varepsilon\sigma$ and still guarantee a uniform lower bound on the contraction rate, with damping effectively scaled by $(1-\varepsilon)$. In practice we compute $\hat\sigma$ using a single JVP along a random direction, update it only every $K$ steps, and stabilise it via EMA and clamping, so moderate noise in high‑dimensional settings is well tolerated; moreover, our LLM experiments estimate $\hat\sigma$ only over LoRA adapter parameters, further reducing variance and cost.
>
>
> ### 5. Could the approach generalize to multi-agent or non-zero-sum games?
>  Our current analysis is carried out for finite two‑player zero‑sum games with an interior Nash equilibrium (Assumptions A1–A3). The entropic Nash mirror ODE and reverse‑KL regularisation themselves extend mechanically to N‑player and general‑sum settings by introducing one simplex and one KL term per agent, and parts of our theory (e.g., the quantal‑response structure and the $O(\beta)$ equilibrium sensitivity via an implicit‑function argument) should carry over with a block‑structured Jacobian. However, the clean spectral separation $\lambda(J_\beta) = -\beta \pm i\sigma_k$​ and the resulting “uniform damping + $\beta$‑invariant rotation” law rely on the two‑player zero‑sum block structure and do not transfer directly to arbitrary multi‑agent/general‑sum games, where additional phenomena can arise. We view a rigorous extension of the $\beta$‑geometry and $\beta$‑controllers to multi‑agent and non‑zero‑sum games as an important and interesting direction for future work.
>
>
> We hope these clarifications and planned revisions address your concerns and better position the contribution as a $\beta$-explicit geometric analysis of reverse-KL Nash mirror dynamics together with practically useful $β$-controllers for stability–bias trade-offs in self-improving LLM agents.

---

### Meta-Review · Area_Chair_ids8 · 2026-01-06

**Summary:**

The paper presents a theoretical analysis of the learning dynamics of self-play trained LLM agents in language game environments.

The reviewers recognized that the analysis is an interesting contribution that advances understanding of learning dynamics of self-play LLM agents (e.g., instability and bias amplification) and offers a way to characterize the tradeoffs. However, the overall sentiment of the reviews remained negative, with the following weaknesses remained unaddressed:
1. As pointed by reviewer JZR6, the manuscript is written in a way that makes it difficult for the reader to follow and understand the core contributions of the work. The authors acknowledged validity of this piece of criticism (in authors' own words: "We appreciate the reviewer’s honest assessment and agree that the paper requires substantial rewriting to be more readable.") but did not submit an updated and revised version of the manuscript.
2. As noted by reviewer 5hAZ, empirical validation of the theory is limited to very small scale / toy experiments. Reviewer BEpr also pointed out that "it feels difficult to evaluate the significance of this contribution without more compelling empirical results."  The authors acknowledge this weakness of their work but chose to not substantiate their results with additional empirical validation further.

Given the overall reviewer sentiment and the unaddressed important concerns, I recommend rejecting the paper. I appreciate the authors actively participating in the discussion and hope it would be useful for them to further improve the paper before resubmitting elsewhere.

**Reviewer Concerns:**

Addressed concerns:
- Answered clarification questions raised by the reviewers.

Unaddressed concerns:
- The manuscript requires a substantial rewrite to improve clarity of presentation.
- The paper can benefit from further empirical validation beyond the tiny scale proof of concept results.

**Reviewer Scores:**

- rzxt: original score of 4, which the reviewer would likely maintain based on the discussion.
- JZR6: original score of 2, which the reviewer would likely maintain based on the discussion.
- BEpr: original score of 4, which the reviewer would likely maintain based on the discussion.
- 5hAZ: original score of 6, which the reviewer would likely maintain based on the discussion.

---

### Decision · Program_Chairs · 2026-01-26

Reject